# Mycotic Keratitis—A Global Threat from the Filamentous Fungi

**DOI:** 10.3390/jof7040273

**Published:** 2021-04-03

**Authors:** Jeremy J. Hoffman, Matthew J. Burton, Astrid Leck

**Affiliations:** 1International Centre for Eye Health, London School of Hygiene and Tropical Medicine, London WC1E 7HT, UK; matthew.burton@lshtm.ac.uk (M.J.B.); astrid.leck@lshtm.ac.uk (A.L.); 2Cornea Service, Sagarmatha Choudhary Eye Hospital, Lahan 56502, Nepal; 3Department of Ophthalmology, Kilimanjaro Christian Medical Centre, P.O. Box 3010, Moshi, Tanzania; 4National Institute for Health Research Biomedical Research Centre for Ophthalmology at Moorfields Eye Hospital NHS Foundation Trust and UCL Institute of Ophthalmology, London EC1V 9EL, UK

**Keywords:** microbial keratitis, fungal keratitis, microbiology, mycotic keratitis, epidemiology, *Fusarium*, *Aspergillus*, dematiaceous fungi, blindness

## Abstract

Mycotic or fungal keratitis (FK) is a sight-threatening disease, caused by infection of the cornea by filamentous fungi or yeasts. In tropical, low and middle-income countries, it accounts for the majority of cases of microbial keratitis (MK). Filamentous fungi, in particular *Fusarium* spp., the aspergilli and dematiaceous fungi, are responsible for the greatest burden of disease. The predominant risk factor for filamentous fungal keratitis is trauma, typically with organic, plant-based material. In developed countries, contact lens wear and related products are frequently implicated as risk factors, and have been linked to global outbreaks of *Fusarium* keratitis in the recent past. In 2020, the incidence of FK was estimated to be over 1 million cases per year, and there is significant geographical variation; accounting for less than 1% of cases of MK in some European countries to over 80% in parts of south and south-east Asia. The proportion of MK cases is inversely correlated to distance from the equator and there is emerging evidence that the incidence of FK may be increasing. Diagnosing FK is challenging; accurate diagnosis relies on reliable microscopy and culture, aided by adjunctive tools such as in vivo confocal microscopy or PCR. Unfortunately, these facilities are infrequently available in areas most in need. Current topical antifungals are not very effective; infections can progress despite prompt treatment. Antifungal drops are often unavailable. When available, natamycin is usually first-line treatment. However, infections may progress to perforation in ~25% of cases. Future work needs to be directed at addressing these challenges and unmet needs. This review discusses the epidemiology, clinical features, diagnosis, management and aetiology of FK.

## 1. Introduction

Mycotic or fungal keratitis (FK) is a severe and potentially blinding infection of the cornea (Figure 1) and is considered an ophthalmic emergency [1,2]. It is one of the leading causes of microbial keratitis (MK) or corneal ulcer. The latest conservative estimates predict that there are close to 1.5 million new infections every year [3], which correlate with estimates published more than 20 years ago [4,5]. The burden of FK is greatest in tropical and subtropical countries, accounting for between 20 and 60% of MK cases presenting in tropical regions [6], likely a result of climate (higher temperatures and relative humidity) and frequent agriculture-related ocular trauma [7].

Fungal keratitis is caused by yeasts and filamentous fungi but the pattern of infection varies globally with respect to aetiology and predisposing risk factors relating to geographical location and occupational exposure. Infections due to *Candida* spp. and other yeasts are typically associated with steroid use, ocular surface disorders, previous ocular surgery, contact lens wear and underlying illness resulting in immuno-incompetency [8], mostly occurring in temperate climes. However, the main burden of disease globally is attributable to the filamentous fungi and these infections predominantly affect the poorest patients in warm, humid, tropical climatic regions [7]. There have also been reports of an increase in *Fusarium*-related keratitis in contact lens wearers in temperate, industrialised regions [9,10,11]. Interestingly, even within developed countries fungal keratitis is a disease of poverty: infections are associated with contact lens wearers from deprived or low socioeconomic backgrounds [3,12].

Over one hundred different species of filamentous fungi isolated from infected corneas have been reported in the literature [13]. The most common genera isolated from filamentous fungal keratitis cases are *Fusarium* spp. and the aspergilli [14], followed by the dematiaceous fungi-a heterogenous group of fungi characterised by melanin-production and pigmentation-*Curvularia* spp. being the most commonly reported genus from this group [15,16,17].

Patients typically present with a red, painful eye, together with reduced vision. Clinical examination will demonstrate conjunctival hyperaemia, making the eye appear red, in conjunction with a corneal infiltrate-an area of corneal opacity, often white or cream in colour (Figure 1A). There will also usually be loss of the corneal epithelium overlying the infiltrate, which stains with topical fluorescein eye drops and fluoresces green under blue light (Figure 1B). These clinical signs can be observed without a slit-lamp, a simple torch with or without loupes will suffice, aided by a blue filter and fluorescein testing strips. More detailed examination using a slit-lamp biomicroscope yields more subtle signs that can help distinguish the different causative agents of MK to some extent; fungal keratitis is more likely if there are serrated margins, raised slough (dead epithelial tissue), and/or colour other than yellow [2].

Unfortunately, presentation to an appropriate eye care provider is usually delayed, with patients often taking a convoluted journey to reach an ophthalmic clinician [18]. Compounding this is the fact that patients often self-medicate with traditional eye medicine which commonly contains non-sterile plant matter, or inappropriate conventional medication (such as topical corticosteroids), exacerbating disease [1,19,20]. Primary health-care workers have little training in recognising, treating or referring MK [21]. This delay leads to advanced infections, which have poor outcomes [1,18]. Accurate diagnosis remains challenging as it is frequently not possible to clinically distinguish bacterial and fungal MK. Microbiology services are usually unavailable. Due to resource limitations in the populations most at risk of these infections microscopy and culture remain the mainstay for diagnosis-Gram, Potassium Hydroxide (KOH), calcofluor white (CFW). Microscopy is the gold standard with visualisation of fungal hyphae in corneal tissue specimens.

Added to this is the fact that FK is particularly challenging to treat. Current topical antifungals are not consistently effective and infections can progress despite prompt treatment, Figure 2 [1,22,23,24,25,26], with up to 30% of patients receiving current ‘gold-standard’ therapy progressing to corneal perforation and/or eye-loss, mediated by human and pathogen derived proteases [1,22,24]. Antifungal drops are rarely available in sub-Saharan Africa and often scarce elsewhere where the burden is greatest [1].

Treatment can be administered topically (first-line, with intensive hourly dosing for at least the first 48 h), orally or by intravenous, subconjunctival, corneal stromal or intracameral injection. The treatment of yeast infections is often different to filamentous fungi, with the former being more common in temperate climates and the latter in hot and humid locations [27,28]. Surgical therapy, typically with therapeutic penetrating keratoplasty (PK, corneal transplantation replacing the diseased cornea with donated corneal tissue), is generally reserved for cases of corneal perforation or progressive infection refractory to medical therapy. PK can also be performed for visual rehabilitation after the acute infection has resolved.

In this review, given the wide range of organisms implicated in fungal keratitis, we have classified the causative organisms into three main groups: hyaline fungi, dematiaceous fungi and yeasts; the focus of this review are the filamentous fungi. The epidemiology, clinical features, diagnosis and management will also be discussed for fungal keratitis as a distinct entity.

## 2. Epidemiology of Fungal Keratitis

### 2.1. Incidence

Until recently, the annual global incidence of fungal keratitis had never been estimated. In 2020, Brown et al. estimated the incidence of fungal keratitis to be 1,051,787 cases per annum, within a range of between 736,251 and 1,367,323 cases per annum [3]. The incidence may in fact be higher at 1,480,916 cases per annum (range 1,036,641–1,925,191) if it is assumed that all unconfirmed culture negative cases of microbial keratitis were in fact fungal in aetiology. The morbidity associated with FK is also important to note: approximately 10–25% of eyes with FK will perforate or need surgical removal, whilst at least 60% of patients, even if treated, are left monocularly blind, equating to approximately 800,000 people per year [1,22,24].

### 2.2. Geographical Distribution

The incidence of FK varies across regions, with the highest incidence in Asia and Africa and the lowest incidence in Europe [3]. Similarly, the proportion of fungal keratitis as a subset of microbial keratitis varies between geographical regions, within the range of 1% of MK cases in Spain to 60% in Vietnam (Table 1) [29,30]. There have been four large reviews that have considered the proportion of MK caused by filamentous fungi by geographical region [3,7,31,32]. The first study from 2002 plotted the proportion of FK as a subset of MK against latitude, and found the proportion of FK cases increases with decreasing latitude, i.e., increasing the closer one is to the equator [7]. The second review, from 2011, correlated the proportion of fungal cases of MK in a country with the country’s gross domestic product (GDP) [31]. This found the highest proportion of fungal infections within Asia, specifically in India and Nepal. The study found the lower the GDP per capita of a country, the higher the proportion of fungal MK. The most recent study looked at both GDP per capita and latitude as potential determinants of the proportion of fungal cases of all those with MK [3]. The findings here correlated to the previous two reviews, suggesting that both proximity to the equator and low GDP per capita are associated with a higher proportion of fungal MK cases [3,7,31]. However, it is important to note there was some considerable unexplained variability [3].

These epidemiological reviews have been updated in Table 1 and Figure 3, which plots the proportion of FK as a subset of MK against distance from the equator [3,32]. There remains a clear inverse correlation with the highest proportion of FK as a subset of MK close to the equator, with the proportion decreasing with increasing distance from the equator. There is, however, considerable variability, and a number of important outliers: Singapore for example is 143 km from the equator but FK accounts for 0.7% of MK cases suggesting that FK is not only climate dependent but also probably linked to rural and occupational risk factors.

In general, filamentous fungal keratitis is a relatively rare cause of MK in temperate regions, where it is often associated with contact lens usage. Table 1 details the proportion of FK (as a subset of MK) in temperate regions such as Europe and temperate North America, which is in the range of 1.2–14.0% [33,34]. This contrasts to tropical regions, such as sub-Saharan Africa and South Asia, where the proportions are considerably higher, with rates reported as between 37.7% and 81.5%, Table 1 [35,36].

**Table 1 jof-07-00273-t001:** Global epidemiology of fungal keratitis (FK), most frequently isolated fungal organisms and summary results of select papers on risk factors for developing FK, grouped as per their geographical region (as defined by the UN).

Country	Year	% FK ^#^	% Culture Negative Cases ^~^	Age (Mean)	% Male	% Trauma	% Steroid	% TEM	% CL	% OSD	% HIV	% DM	Distance from Equator (km)	N (All MK Cases)	Organism 1 (%)	Organism 2 (%)	Reference
**AFRICA**																	
Egypt (Zagazig)	2012	55	-	-	-	63.6	6.1	-	12.1	-	-	18.2	3401	60	*Penicillium* spp. (24.2)	*Aspergillus* spp. (21.2)	[37]
Egypt (Mansoura)	2013–2015	65.5	55.5	-	66.1	51.4	5.3	-	2.4	4.5	-	15.1	3452	247	*Aspergillus* (41.0)	-	[38]
Egypt (Tanta)	2011–2013	43.3	15.5	49.2	65.9	58.4	32.7	-	-	-	-	-	3424	834	*Aspergillus flavus* (29.1)	*Aspergillus niger* (16.1)	[39]
Ethiopia	2014–2015	45.1	-	-	58.0	78.3	5.8	-	-	-	-	7.2	1003	153	*Fusarium* spp. (27.6)	*Aspergillus* spp. (25.0)	[40]
Ghana	1995	56.1	42.7	36.3	69.3	-	-	-	-	-	-	-	900	199	*Fusarium* spp. (52.3)	*Aspergillus* spp. (15.3)	[41]
Ghana	1999-2001	74.7	60.0	-	-	-	-	-	-	-	-	-	900	290	*Fusarium* spp. (42.2)	*Aspergillus* spp. (17.4)	[7]
Libya	2008–2010	32.9	^	-	60.7	78.5	3.6	-	21.4	-	-	17.8	3114	85	*Aspergillus* spp. (50.0)	*Fusarium* spp. (39.3)	[42]
Sierra Leone	2005–2006	37.7	5.5	-	-	-	-	-	-	-	-	-	941	73	UFF (69.2)	*Aspergillus* spp. (15.4)	[35]
South Africa	1982-1983	3.7	-	-	-	0	0	-	0	-	33.3	50.0	2913	164	*Curvularia* spp. (33.3)	-	[43]
South Africa	2013–2015	2.2	-	-	-	-	-	-	-	-	-	-	3399	46	-	-	[44]
Tanzania	2008–2010	51.6	45.6	-	-	-	-	-	-	-	-	-	667	170	UFF (87.5)	*Candida* spp. (12.5)	[1]
Tanzania	2013	40.2	8.9	-	-	-	-	-	-	-	-	-	755	202	*Candida* spp. (60.8)	UFF (39.6)	[45]
Tunisia	1995-2012	12.4	7	47.2	63.3	61.6	18.3	3.3	3.3	10.0	-	5	3862	483	*Fusarium* spp. (49.0)	*Aspergillus* spp. (22.0)	[46]
Tunisia	2010–2015	30.0	40	48.9	56.6	43.3	3.3	-	3.3	-	-	10	4124	30	*Fusarium* spp. (50.0)	*Aspergillus* spp. (33.3)	[47]
Tunisia	1996–2004	21.6	58	-	-	50.0	-	-	-	25	-	-	4094	100	*Fusarium* spp. (87.5)	*Acremonium* spp. (12.5)	[48]
**ASIA**																	
Bangladesh	2008	39.2	5	-	-	49	-	-	-	-	-	-	2752	120	-	-	[49]
Bangladesh	1994	36	18.3	-	-	-	-	-	-	-	-	-	2483	142	*Aspergillus* spp. (40.0)	*Fusarium* spp. (21.0)	[50]
China	2009–2013	45.7	53.9	-	-	-	-	-	-	-	-	-	2504	2973	*Fusarium* spp. (29.3)	*Aspergillus* spp. (24.1)	[51]
China	1999-2004	61.9	-	-	-	-	-	-	-	-	-	-	4012	1054	*Fusarium* spp. (73.3)	*Aspergillus* spp. (12.1)	[52]
China (Hong Kong)	1997-1998	6.4	65.0	-	-	20	20	-	-	-	-	-	2491	223	*Fusarium* spp. (60.0)	*Penicillium* spp. (20.0)	[53]
China (Hong Kong)	2004–2013	10.7	67.7	-	-	-	-	-	-	-	-	-	2491	260	*Fusarium* spp. (33.3)	*Candida* spp. (25.0)	[54]
China (Taiwan)	1992-2001	13.5	51.0	-	-	-	-	-	-	-	-	-	2758	453	*Fusarium* spp. (29.4)	*Candida* spp. (29.4)	[55]
China (Taiwan)	2012–2014	8.2	69	-	-	-	-	-	-	-	-	-	2669	233	*-*	-	[56]
India (West Bengal)	2007–2011	58.9	27.0	-	61.7	88.7	16.3	-	-	6.0	-	11.8	2510	928	*Aspergillus* spp. (37.8)	*Fusarium* spp. (20.3)	[57]
India (West Bengal)	2008	38.1	32.7	53	65.0	48.0	16.0	16.0	-	-	-	-	2510	289	*Aspergillus* spp. (55.4)	*Fusarium* spp. (10.8)	[58]
India (Odisha)	2006–2009	35.5	18.6	-	70.0	40.2	-	-	-	-	-	2.2	2253	997	*Aspergillus* spp. (27.9)	*Fusarium* spp. (23.2)	[59]
India (Assam)	2007–2009	60.6	-	-	68.8	76.40	-	-	-	1.5	-	2.5	3056	310	*Fusarium* spp. (25.0)	*Aspergillus* spp. (19.0)	[60]
India (Delhi)	2010–2015	68	65	-	76	89.4	-	-	-	5.3	-	1.3	3188	400	*Aspergillus* spp. (30.8)	*Fusarium* spp. (27.6)	[61]
India (Chandigarh)	2005–2011	Ψ	49	-	-	66.5	2	-	-	11.7	-	-	3418	765	*Aspergillus* spp. (47.6)	Dematiaceous fungi (21.9)	[62]
India (Rajasthan)	2005–2012	68.2	45	-	71.7	62.8	-	-	3.9	-	1.1	8.9	3116	480	*Aspergillus* spp. (63.3)	*Alternaria* spp. (8.3)	[63]
India (Chandigarh)	1999-2003	41.5	46.9	-	80.5	43.8	7.8	-	-	-	-	-	3418	64	*Aspergillus* spp. (41.2)	*Fusarium* spp. (23.5)	[64]
India (Delhi)	2007–2011	58.9	27	-	61.7	88.7	16.3	-	-	19	-	11.8	3188	928	*Aspergillus* spp. (37.8)	Dematiaceous fungi (23.8)	[65]
India (Delhi)	2000–2004	22.3	-	-	77.9	32.4	16.2	2.7	0	-	-	-	3188	346	*Aspergillus* spp. (55.9)	Dematiaceous fungi (7.8)	[66]
India (Madurai)	2012–2013	79	14.5	50	64	70	9	19				8	1103	252	*Fusarium* spp. (39.0)	*Aspergillus* spp. (18.0)	[67]
India (Madurai)	1999–2002	52.1	29.4	-	65	92.1	1.2	-	0	6.7	-	15.7	1103	3183	*Fusarium* spp. (41.9)	Dematiaceous fungi (26.9)	[68]
India (Madurai)	1994	51.8	31.6	-	61.3	-	-	-	-	-	-	-	1103	434	*Fusarium* spp. (47.1)	*Aspergillus* spp. (16.1)	[4]
India (Hyderabad)	1991–2000	39.8	^	40.4	71.2	54.4	5.9	-	-	11.7	-	6.4	1933	3399	*Fusarium* spp. (37.2)	*Aspergillus* spp. (30.7)	[69]
India (Hyderabad)	1991–2001	44.8	39.6	30.9	-	81.9	2.4	-	0.3	18.2	-	-	1933	5897	*Fusarium* spp. (35.6)	*Aspergillus* spp. (26.8)	[70]
India (Madurai)	2006–2009	63	42	-	-	-	-	-	-	-	-	-	1103	6967	*Fusarium* spp. (42.3)	-	[71]
India (Bangalore)	2012–2014	55.5	62.5	-	-	-	-	-	-	-	-		1442	312	*Fusarium* spp. (31.0)	*Aspergillus* spp. (11.0)	[72]
India (Tamil Nadu)	1999–2001	44	31	-	-	-	-	-	-	-	-	-	1223	800	*Aspergillus* spp. (39.9)	*Fusarium* spp. (21.5)	[7]
India (Maharashtra)	2004–2009	57.9	37	-	-	-	-	-	-	-	-	-	2120	852	*Fusarium* spp. (35.0)	*Aspergillus* spp. (18.0)	[12]
India (Gujarat)	2006–2008	65	60		54	73	-	-	-	-	-	-	2498	100	*Aspergillus* spp. (70.0)	*Fusarium* spp. (12.0)	[73]
India (Gujarat)	2003–2005	51.8	45	-	-	-	-	-	-	-	-	-	2498	200	*Fusarium* spp. (29.8)	*Aspergillus* spp. (21.1)	[74]
India (Gujarat)	2007–2008	34.9	40.7	-	-	-	-	-	-	-	-	-	2561	150	*Aspergillus* spp. (35.4)	*Fusarium* spp. (22.5)	[75]
India (West Bengal)	2001–2003	62.7	32	-	-	-	-	-	-	-	-	-	2669	1198	*Aspergillus* spp. (59.9)	*Fusarium* spp. (21.2)	[76]
India (Delhi)	2005	49.1	43.2	-	-	-	-	-	-	-	-	-	3183	1000	*Aspergillus* spp. (41.6)	*Fusarium* spp. (19.8)	[77]
India (Hyderabad)	2002	19.4	33.5	-	-	-	-	-	-	-	-	-	1933	170	*Fusarium* spp. (72.7)	-	[78]
Iran (Tehran)	2011–2013	Ψ	94.4		79.3	-	-	-	-	-	-	-	3969	2180	*Fusarium* spp. (49.6)	*Aspergillus* spp. (26.4)	[79]
Iran (Sari)	2004–2005	77.8	59.1	61.5	71.4	28.6	0	-	0	14.3	-	14.3	4065	22	*Fusarium* spp. (50.0)	*Aspergillus* spp. (50.0)	[80]
Iraq	2002–2005	31.9	41.4	-	-	90	-	-	0	6.8	-	-	3707	396	*Aspergillus* spp. (56.8)	*Fusarium* spp. (27.0)	[81]
Iraq	2013–2014	6.8	30.5	-	-	-	-	-	-	-	-	-	3707	105	*Aspergillus* spp. (60.0)	*Alternaria* spp. (40.0)	[82]
Iraq	2017–2018	37	^	73	61	-	-	-	-	-	-	41	3707	234	*Aspergillus* spp. (70.0)	*Penicillium* spp. (13.0)	[83]
Japan	1999–2003	6.1	41.5	-	-	-	-	-	-	-	-	-	3991	122	*Candida* spp. (83.3)	-	[84]
Japan	2003	10.6	56.7	-	-	-	-	-	-	-	-	-	3969	261	-	-	[85]
Korea (RO)	2003–2008	26.9	37.3	-	-	-	-	-	-	-	-	-	4177	83	*Candida* spp. (57.0)	*Aspergillus* spp. (28.6)	[86]
Malaysia	2007–2011	25.3	12.8	-	61.7	48.9	17.0	-	4.3	10.6	-	10.6	371	186	*Fusarium* spp. (46.0)	*Aspergillus* spp. (9.8)	[87]
Malaysia	2017	36.4	59.9	-	-	-	-	-	-	-	-	-	367	137	*Fusarium* spp. (60.0)	-	[88]
Nepal (Dharan)	2004–2008	61.1	20.8	-	-	-	-	-	-				2980	351	*Aspergillus* spp. (33.3)	*Fusarium* spp. (12.7)	[89]
Nepal (Dharan)	1998–1999	65.5	32.6	-	-	-	-	-	-	-	-	-	2980	86	*Aspergillus* spp. (60.5)	*Fusarium* spp. (13.2)	[90]
Nepal (Nepalgunj)	2011–2012	36	^	-	59.3	58	12	-	-	6	-	-	3120	1880	*Fusarium* spp. (31.9)	*Curvularia* spp. (17.7)	[91]
Nepal (Dharan)	2007–2008	60	54.5	-	-	-	-	-	-	-	-	-	2980	44	*Aspergillus* spp. (66.6)	-	[92]
Nepal (Kathmandu)	2014	44	55.4	-	-	-	-	-	-	-	-	-	3080	101	*Fusarium* spp. (24.0)	*Aspergillus* spp. (20.0)	[93]
Nepal (Kathmandu)	1981	25	50	-	-	-	-	-	-	-	-	-	3080	133	-	-	[94]
Nepal (Biratnagar)	2011	*70*		-	-	-	-	-	-	-	-	-	2944	1644	No culture performed, microscopy only		[95]
Oman	2004–2007	31.3	57.9	-	59.4	25	31.3	15.6	-	18.8	-	9.4	2510	242	*Fusarium* spp. (50.0)	*Aspergillus* spp. (34.4)	[96]
Oman	2000–2006	11.8	56.9	-	-	-	-	-	-	-	-	-	2510	188	-	-	[97]
Pakistan	2010	64	32.3	-	-	-	-	-	-	-	-	-	2788	133	-	-	[98]
Saudi Arabia	1984–2004	10.3	69.4	55	79	20.9	16.9	-	0.8	8.87	-	12	2746	1200	*Aspergillus* spp. (37.0)	*Trichophyton* spp. (20.0)	[99]
Singapore	1991–2005	Ψ	^	40	79.3	55	24	-	7	14	-	-	143	29	*Fusarium* spp. (52.0)	*Aspergillus* spp. (17.0)	[100]
Singapore	2012–2014	0.7	-	-	-	-	-	-	-	-	-	-	143	531	-	-	[56]
Sri Lanka	1976–1981	81.5	59.1	-	-	-	-	-	-	-	-	-	811	66	UFF (63.6)	*Aspergillus* spp. (18.0)	[36]
Thailand (Central)	1988–2000	24.6	52.7	-	-	-	-	-	-	-	-	-	1529	292	*Fusarium* spp. (34.3)	*Aspergillus* spp. (20.0)	[101]
Thailand (South)	1982–2003	15.3	-	46.4	72.3	66	-	-	-	-	-	-	800	556	*Fusarium* spp. (64.5)	*Aspergillus* spp. (10.5)	[102]
Thailand (North)	2003–2006	50.8	74.4	-	-	-	-	-	-	-	-	-	2090	305	*Fusarium* spp. (58.1)	*Aspergillus* spp. (12.9)	[103]
Thailand (Central)	2001–2004	38	^	-	67.7	77.5	-	-	0	9.68	-	-	1529	127	*Fusarium* spp. (26.0)	Dematiaceous fungi (20.0)	[104]
Turkey (Adana)	2014–2015	9.4	-	39.3	50	50	-	-	25	-	-	-	4115	64	*Aspergillus* spp. (66.7)	*Fusarium* spp. (33.3)	[105]
Turkey (West Anatolia)	1990–2005	22.5	63.8	-	-	-	-	-	-	-	-	-	4278	620	*Fusarium* spp. (50.0)	*Aspergillus* spp. (20.0)	[106]
Vietnam (North)	2008	59.6	47.2	-	44.1	83.8	1.4	1.4	-	-	-	-	2338	1153	*Fusarium* spp. (40.7)	*Aspergillus* spp. (25.9)	[30]
Vietnam	1974–1982	23.6	-	-	-	-	-	-	-	-	-	-	2338	1219	-	-	[107]
**EUROPE**																	
Netherlands	2002–2004	1.8	42.0	-	-	-	-	-	-	50	50	-	5823	156	*Candida albicans* (100)	-	[108]
Netherlands	2014–2017	14.0	50	-	-	-	-	-	-	-	-	-	5809	185	-	-	[34]
UK (SW England)	2006–2017	6.9	61.9	-	-	-	-	-	-	-	-	-	5721	2116	UFF (54.2)	*Candida* spp. (45.8)	[109]
UK (London)	2007–2014	-	34.8	47.2	41.4	11.6	32.1	-	57.1	22.3	-	-	5727	112	*Fusarium* spp. (41.8)	*Candida* spp. (38.0)	[28]
UK (NE England)	2008–2017	4.2	55.5	55.3	65	-	-	-	-	-	-	-	6113	407	UFF (50.0)	*Candida* spp. (50.0)	[110]
UK – (NW England)	2004–2015	7.1	67.4	-	-	-	-	-	-	-	-	-	5980	4229	*Candida* spp. (53.2)	*Fusarium* spp. (25.7)	[111]
**LATIN AMERICA AND THE CARRIBEAN**																	
Brazil (São Paulo)	1975–2007	11	51.4	-	-	-	-	-	-	-	-	-	2547	6804	*Fusarium* spp. (51.9)	*Candida* spp. (17.6)	[112]
Brazil (Uberlandia)	2001–2004	56.3	50.8	-	-	55.6	-	-	0	0	-	-	2104	65	*Fusarium* spp. (61.1)	*Aspergillus* spp. (16.7)	[113]
Brazil (São Paulo)	2000–2004	13.8	63.4	40.7	80.3	-	-	-	-	-	-	-	2547	478	*Fusarium* spp. (66.7)	*Aspergillus* spp. (10.6)	[114]
Brazil (São Paulo)	2003–2010	25	82.4	43	74	49.3	-	-	-	-	-	-	2547	599	*Fusarium* spp. (83.3)	*Aspergillus* spp. (16.7)	[115]
Mexico	2013–2014	33.3	47.1	-	-	-	-	-	-	-	-	-	2161	51	*Fusarium* spp. (44.4)	*Aspergillus* spp. (22.2)	[116]
Paraguay	1988–2001	49	21	-	-	-	-	-	-	-	-	-	2814	660	*Acremonium* spp. (40.0)	*Fusarium* spp. (15.0)	[117]
Paraguay	2009–2011	72.1	10.4	-	71	-	-	-	-	-	-	-	2814	48	*Fusarium* spp. (34.0)	*Aspergillus* spp. (16.1)	[118]
**NORTH AMERICA**																	
USA (N California)	1976–1999	8.4	62	-	-	-	-	-	-	-	-	-	4201	1121	*Candida* spp. (30.5)	-	[119]
USA (Florida)	1968–1977	35.8	44.0	-	-	-	-	-	-	-	-	-	2865	663	*Fusarium* spp. (62.0)	*Candida* spp. (7.5%)	[120]
USA (Florida)	1999–2006	-	29.8	48	75	43	29	-	44	8.3	-	7.1	3298	84	*Fusarium* spp. (41.0)	*Candida* spp. (14.0)	[121]
USA (S California)	1998–2008	1.4	^	56.1	54	14	-	-	24	12.7	1.6	16	3638	4651	UFF (64.0)	*Candida* spp. (32.0)	[122]
USA (New York)	1987–2003	1.2	^	47	35	11	7	-	10	23	25	7	4528	5083	*Candida* spp. (66.0)	*Aspergillus* spp. (12.0)	[33]
**OCEANIA**																	
Australia (Brisbane)	1999–2004	8	35	-	-	-	-	-	-	-	-	-	3054	231	*Fusarium* spp. most commonly isolated	-	[123]
Australia (Queensland)	1996–2016	-	^	48	65	-	-	-	-	-	-	-	3054	215	*Fusarium* spp. (33.3)	*Aspergillus* spp. (13.0)	[124]
Australia (Sydney)	2009–2017	-	6	60	65	24	54	-	26	34	-	-	3764	51	*Candida* spp. (33.0)	*Fusarium* spp. (28.0)	[125]
Australia (Queensland)	2005–2015	6	^	-	-	-	-	-	-	-	-	-	3054	3182	UFF (75.9)	*Candida* spp. (24.1)	[126]
Australia (Sydney)	2012–2016	3.3	31	63.5	67	25	46	-	28	25	-	8	3764	1052	*Candida* spp. (30.4)	*Fusarium* spp. (21.7)	[127]
New Zealand	2003–2007	1.7	34.4	-	-	-	-	-	-	-	-	-	4097	265	*Fusarium* spp. (66.7)	*Candida* spp. (33.3)	[128]

^#^ Confirmed fungal keratitis cases as a percentage of all culture positive microbial keratitis cases, including mixed bacterial-fungal infections. If diagnosis was based on microscopy (culture unavailable), this is a percent of all microbial keratitis cases examined by microscopy, and the results of these are given in italics. ^~^ Culture negative rate of all cultures taken within the study. - Data not presented. Ψ Studies that only included cases of FK and did not report the number of MK cases. ^ Studies that only included cases that were culture positive and did not report the overall culture negative rate. FK, fungal keratitis; TEM, traditional eye medication; CL, contact lens; OSD, ocular surface disease; HIV, Human Immunodeficiency Virus; DM, diabetes mellitus; MK, microbial keratitis; UFF, unspecified filamentous fungi.

Globally, *Fusarium* spp. and *Aspergillus* spp. are the most commonly isolated fungal causes of FK and are discussed in more detail below. Note, however, that non-filamentous FK is generally more common in temperate climates, where *Candida* spp. is most frequently implicated.

### 2.3. Changing Incidence over Time

There is evidence that the proportion of MK attributable to fungi is increasing over time, particularly in low and middle-income countries (LMICs) [3]. For example, in Thailand between 1982 and 2003, the mean proportion of FK cases was 13.6% [102]. This increased to 50.8% between 2003 and 2006 [103]. Similar increases have been observed in other parts of Asia, including Nepal with an increase form 23.1% in 1981 to 70% in 2011 [94,95]. Increases have also been observed in Africa, for example in Ghana where the percentage of FK cases increased from 56.1% in 1995 to 74.7% between 1999 and 2001 [7,41]. For countries where there are multiple reports published at different time-points which we reviewed in Brown et al. [3], the relative proportion of FK is plotted against time in Figure 4.

The reason for the increase in LMICs is unclear and has not been formally studied. It could be attributable to the increased use of topical antibiotics as a primary prevention measure following corneal abrasions or as empirical treatment at a primary health level for microbial keratitis, resulting in only severe or resistant bacterial infections presenting to secondary or tertiary care along with all fungal cases. It may also be driven due to greater availability of topical antibiotics available without prescription from pharmacies. Another potential reason for this increase may include climate change: a study from Egypt in 2011 found a strong correlation between the increase in cases of fungal keratitis between 1997 and 2007 and the increase atmospheric temperature and humidity detected during the same period [129]. Other potential reasons include increased availability and use of topical steroids, increased prevalence of diabetes mellitus across the regions or simply due to improved culture and microbiology services in these countries, meaning that under-reported previous incidence is now being reported more accurately. Increased contact lens wear may also be a contributing factor, although on the whole contact-lens use remains infrequent in poorer countries across Asia and Africa.

In developed countries, there is also evidence of an increasing incidence over time, attributed to the widespread use of contact lenses, including bandage contact lenses, as well as topical steroid use [121,130]. For example, a study from a tertiary referral hospital in Florida showed an increase of over 100% in the number of cases of fungal keratitis between 1999 and 2005; contact lens wear was found to be the most common risk factor in this study [121]. A retrospective multi-centre case series from the US reported a significant increase in the incidence of non-*Fusarium* filamentous fungal keratitis cases between the period 2001–2004 and 2004–2007 (*p* < 0.0001) [130]. The number of *Fusarium* cases increased substantially between 2004 and 2006, when ReNu with MoistureLoc contact lens cleaning solution was on the market, and then returned to the pre-2004 incidence level for the remainder of the study [130]. For the increase described for non-*Fusarium* cases, the authors were unable to give a clear reason why this may have occurred; a contact lens-related product was unlikely to be responsible as the similar trends were seen for both contact lens wearers and non-contact lens wearers [130]. A more recent study from a tertiary eye hospital in the UK also reported a significant increase in the number of cases of filamentous fungal keratitis between 2007 and 2014 (*p* = 0.005), whilst there was no significant change in the incidence of yeast infections (*p* = 0.3) [28]. The same study also compared the incidence between data collected between 1994 and 2006 and data from 2007 to 2014, and found a significant increase in fungal keratitis cases (*p* = 0.03) [28]. All three of these studies report an increasing proportion of filamentous FK compared to yeast FK, with filamentous organisms (and in particular *Fusarium* spp.) now responsible for the majority of FK cases [28,121,130]. More research is required through case–control or national surveillance studies to explore reasons behind this apparent increase in incidence over time in both temperate and tropical locations.

### 2.4. Risk Factors

There are numerous risk factors for developing fungal keratitis, some attributable to the individual such as age, gender or pre-existing ophthalmic or systemic disease, with others dependent on extrinsic factors including the income status of the patient, occupation, contact-lens use, previous ocular surgery and region. Select risk factors from a number of epidemiological studies on fungal keratitis are presented in Table 1.

#### 2.4.1. Age and Gender

Despite age and gender not being independent risk factors for fungal keratitis, they both affect other risk factors such as trauma, which is more common in younger men who tend to be agricultural labourers [12,131]. It is also important to note that older patients tend to have a more severe disease and worse outcome [108]. Furthermore, older patients are more likely to have predisposing systemic and ocular co-morbidities such as diabetes mellitus and ocular surface disease [108]. Patients between the ages of 20–40 make up the majority of cases [12,28,69,131]. In areas of high incidence of fungal keratitis such as south India, the majority of young patients (aged between 21 and 50) typically have fungal keratitis, compared to the majority of patients over 50 years old who typically have bacterial keratitis [68].

In SSA and India where the burden of FK is greatest, the majority of cases of fungal keratitis are reported in males [12,40,69]. Interestingly, one study from Nepal reports a higher proportion of females compared to males [93], whilst other studies from Nepal report male preponderance [91,95]. The reason for this difference is unclear; it may be due to different socioeconomic factors, health seeking behaviour or differing study methodology. In Europe and North America, there is considerable variation in the reported proportion of men with fungal keratitis [28,121].

#### 2.4.2. Trauma

Preceding ocular trauma is a key predisposing risk factor for the development of fungal keratitis, regardless of geographical region. This is particularly true for trauma with vegetative material and trauma occurring during agricultural practices. Injury to the eye allows for a disruption to the corneal epithelium, permitting fungal pathogens to infiltrate the cornea [24,46,68,132,133,134]. Furthermore, injury with plant matter can lead to direct inoculation with fungal conidia. For regions where a fungal aetiology is the most common form of microbial keratitis such as South Asia and SSA, the reported rates of trauma range from 24 to 83% [1,76].

#### 2.4.3. Occupation

Given the clear risk that trauma, particularly with organic material, poses to the cornea it is not surprising that occupations that carry a high risk of occupational ocular injury are associated with developing fungal keratitis. In particular, agricultural labourers and subsistence farmers are the most likely to develop fungal keratitis, reported to be between 56–74% of cases from studies in Nepal and India [12,91].

#### 2.4.4. Diabetes Mellitus

Diabetes mellitus (DM) is of increasing public health concern globally, with the incidence increasing at an alarming rate in LMICs [135]. It is well-established that patients with DM are at an elevated risk of developing fungal infections [136], and DM is the most important systemic risk factor for developing fungal keratitis [60]. DM has also been shown to be an independent risk factor for the severity of fungal keratitis [137]. It is thought that hyperglycaemia can alter the ocular surface microenvironment including changes to the commensal organisms and enzyme action, allowing easier fungal adherence, proliferation and corneal penetration [137]. The associated reduced immune response seen in diabetes is also likely to be a significant factor in increasing host susceptibility to fungal infection [138].

#### 2.4.5. HIV

There have been a number of studies from SSA that have suggested an association between HIV infection and fungal keratitis, following a number of case reports of fungal keratitis in AIDS patients at the start of the HIV/AIDS pandemic [139,140]. A prospective study from Tanzania in 1999 found that 81% of the patients with fungal keratitis were HIV positive, compared to only 33% in non-fungal cases (*p* < 0.001) [141]. Another study from Tanzania a few years later found the prevalence of HIV infection amongst MK cases to be double that of the wider population [1], although this did not directly compare the proportion of HIV positive fungal MK cases to bacterial MK cases. A more recent, nested case control study from Uganda where over 60% of MK cases were fungal, found a strong association between HIV infection and MK (OR 83.5, *p* = 0.02) [138,142]. There have been no studies to date looking at this association outside of SSA. 

#### 2.4.6. Traditional Eye Medicine

The use of traditional eye medicine (TEM) to treat a wide range of eye problems is commonplace in LMICs [143,144]. Most TEM contain non-sterile preparations comprising plant matter, often herbs or dried leaves, and are therefore a potential route for inoculating the cornea with microorganisms, particularly fungal pathogens [60]. Although there are no studies that have specifically looked at TEM as a risk factor for fungal keratitis, it has been found to be an independent risk factor in developing microbial keratitis in Tanzania and Uganda [20,138,142], where a fungal aetiology make up the majority of MK cases.

#### 2.4.7. Topical Corticosteroids

It is well established that glucocorticoids are associated with an increased risk of invasive fungal infection due to the dysregulation of the patient’s immunity [145]. This holds true for prior topical steroid use, which is an independent risk factor for developing fungal keratitis [146]. Prior topical corticosteroid use is also associated with deeper corneal penetration and a worse clinical outcome [147]. Although topical corticosteroid use is associated with both yeast and filamentous fungal infections, it may be a stronger risk factor for yeast infection [28].

#### 2.4.8. Ocular Surface Disease

Pre-existing ocular surface disease (OSD, a diverse range of disorders that lead to an abnormal ocular surface such as dry eye disease, corneal exposure, blepharitis, persistent epithelial defects or ocular surface inflammatory conditions) compromises the corneal epithelium and therefore allows fungal pathogens to invade the cornea. Furthermore, these conditions are often treated with topical corticosteroids or bandage contact lenses, which further increases the risk of developing fungal keratitis. Although OSD is more often associated with yeast infection [28], it remains a risk factor for filamentous fungal infection: a multi-centre study from the US found 29% of cases of fungal keratitis were associated with OSD, 42.6% of which were filamentous and 53.1% were yeast [148]. Cases of fungal keratitis with pre-existing OSD are less frequently reported in LMICs than in developed countries, other than in areas such as Tanzania, where OSD due to trachoma exists [149].

#### 2.4.9. Contact Lens Usage

In industrialised countries, contact lens use constitutes the main predisposing factor for developing fungal keratitis, with studies showing between 37% and 67% of fungal cases were contact lens wearers [28,130,148]. It is important to consider, however, that it is not simply the contact lens wear itself that carries the risk-it is the type of lens used, the frequency of replacement and how the lenses are cleaned-and with what. For example, the global outbreak of *Fusarium* keratitis between 2005–2006 was caused by a specific contact lens cleaning solution [150]. The current proportion of patients with fungal keratitis in LMICs associated with contact lens usage is low, but this is likely to increase as these countries industrialise leading to an increased number of contact lens wearers and fewer people involved in manual agricultural labour.

#### 2.4.10. Previous Ocular Surgery

A prior history of ocular surgery, including cataract, laser-refractive or corneal transplantation surgery, has been associated with the development of fungal keratitis in both developed and lower-middle income countries [151,152]. Yeasts are often the most commonly implicated pathogen following surgery [8]; for example, in a study from Boston, USA, yeasts accounted for 67% of post-surgical fungal infections. Of note, this group of patients had the worst outcome in terms of final visual acuity. In this study, all surgeries were a form of corneal transplantation [153]. However, it should be noted that prior ocular surgery is more likely to be a stronger risk factor for bacterial, rather than fungal, keratitis [61]; a study from Brazil found 32% of bacterial keratitis cases were associated with previous ocular surgery, compared to just 8% of fungal keratitis cases [154].

Despite intravitreal injections for retinal disease becoming the most commonly performed intraocular procedure globally [155], and corticosteroid periocular injections being used routinely for the treatment of diabetic macular oedema [156,157], there have been no cases of fungal keratitis associated with this treatment reported in the scientific literature to date. However, other complicating local fungal infections have been reported, including fungal endophthalmitis, fungal orbital abscesses and conjunctival mycetoma [158,159,160].

## 3. Clinical Features

It can be challenging to distinguish fungal keratitis from other forms of microbial keratitis, and even more difficult to distinguish different fungal aetiologies on clinical grounds. For example, a study whereby fifteen ophthalmologists had to predict the likely microbiological aetiology found that fungal keratitis was the most challenging to diagnose, with a sensitivity and specificity of 38% and 45%, respectively [161], whilst in a separate study using corneal photographs, corneal specialists were only able to correctly differentiate fungal and bacterial keratitis in 66% of cases [162].

There are, however, some clinical signs that have been shown to be useful predictors for filamentous fungal keratitis [2]. These are serrated margin, raised slough and colouration other than yellow. If one of these signs was present, the probability of fungal infection was 63%; if more than one of these were present the probability was 83% [2]. Without using colour as a discriminator, the probability increased to 89% [163]. Satellite lesions, which have historically been believed to be discriminatory for fungal keratitis, have been shown to occur in *Acanthamoeba* and fungal keratitis with the same frequency and are no more frequent in fungal than bacterial keratitis [164].

Some clinical features have been found to be more likely associated with *Fusarium* infection compared to *Aspergillus* infection. For example, *Fusarium* ulcers are more likely to have serrated (or “feathery”, indistinct) margins or edges and non-yellow infiltrate (Figure 5A), whilst cases of *Aspergillus* keratitis are more likely to have a raised surface or presence of hypopyon (Figure 5B) [67]. Another study agreed with these findings, with *Aspergillus* cases more likely to have a raised surface, but also presence of an endothelial plaque; these were less common in *Fusarium* cases [165]. Ring infiltrates were also predictive of *Aspergillus.* Pigmented corneal infiltrates are very likely to be caused by dematiaceous fungi; in the study by Oldenburg et al. all pigmented corneal ulcers were dematiaceous [165]. Presence of a raised profile is also associated with dematiaceous fungi such as *Curvularia* spp. (Figure 5C) [15,165,166].

Despite the above clinical signs being more frequently associated with fungal keratitis, other studies have shown a lack of statistical significance [161,164]. This adds to the challenge to accurately and confidently diagnose fungal keratitis on clinical grounds alone. Compounding this is the pleomorphic presentation as a result of late presentation, prior use of topical steroids or traditional eye remedies that unfortunately often occurs frequently in the regions where fungal keratitis is most prevalent [18].

Acutely, fungal keratitis typically leads to reduced vision due to the presence of the infection and inflammation in the cornea, blurring the vision. With treatment, the vision can improve, although often the patient is left with worse vision than they had previously due to the development of corneal scarring. At present there is no medical treatment to reverse this scarring process. Rigid contact lenses can help to a certain amount by improving the vision if there is scarring. Alternative options for severe scarring include corneal graft surgery, but this can be a technically challenging procedure and is often not available in places most in need. Fungal keratitis should therefore be considered a potentially blinding condition.

## 4. Making the Diagnosis

Even with all diagnostic modalities available, diagnosing fungal keratitis can be challenging. The burden of fungal keratitis globally is predominantly in low resource settings, where access to advanced diagnostic techniques is very limited. In these locations, diagnoses are still often made on clinical grounds alone (with the associated limitations as discussed above), sometimes supported by basic microscopy. However, an algorithm has been developed that uses the specific features that were systematically examined from a large case series from Ghana [163], and calculating a probability score that the microbial keratitis is fungal in aetiology, Figure 6 [2]. This can aid clinicians working in these locations and indicate the likelihood of fungal versus bacterial infection. Where diagnostic microbiology is available, however, it is best practice to rely on the results of this rather than these clinical signs, as the presence of fungal hyphae in corneal tissue is diagnostic [163].

### 4.1. Laboratory

#### 4.1.1. Microscopy and Culture

Infected corneal tissue/material is gently removed from the surface of the anaesthetised cornea using a sterile needle or scalpel blade and transferred to microscope slides and a range of solid and liquid phase culture media, including blood agars and Sabouraud dextrose agar [13].

Microscopy is still regarded as the gold standard in laboratory diagnosis of fungal keratitis and is often the only diagnostic tool available in settings where the incidence of FK is highest. The presence of fungal hyphae in corneal scrape preparations is always significant and are clearly visible using Gram stain, KOH, CFW or lactophenol cotton blue (LPCB, Figure 7) [13,167]. The ubiquitous distribution and environmental reservoirs of fungal ocular pathogens mean that positive microscopy is critical to exclude contaminants.

Culture positivity rates reported vary greatly between institutions and settings [32]. Low culture positivity is attributable to the very small size of the specimen, use of antimicrobial agents by the patient prior to presentation, the quality of the corneal scrape and incorrect inoculation of media, in addition to laboratory factors [168,169]. Subculture for identification to species level may require the use of plant-based agars, most commonly examples are potato dextrose and cornmeal agars, in addition to diurnal culture methods to induce sporulation for the purpose of identification.

#### 4.1.2. Molecular Techniques

Rapid diagnosis to inform prompt and appropriate treatment is critical to the successful clinical management of fungal keratitis. Development of molecular techniques, such as pan-fungal 16S rRNA PCR, have been favoured due to the very small size of specimen. PCR has emerged as both sensitive and specific test for the diagnosing fungal keratitis, benefiting from a high positive detection rate [14,170,171,172,173,174], with some evidence that it may be more sensitive than the traditional microbiological techniques of microscopy and culture [175]. However, the accuracy of PCR to diagnose fungal keratitis is dependent on adequate sampling and the primers used. Recent promising developments include evaluation of ITS primers and multiplex PCR for direct identification of fungal species from corneal tissue demonstrating high sensitivity and specificity [14].

An area currently under research that could have important therapeutic and prognostic implications is the development of genotyping methods for rapid species identification. This has shown promise for the rapid detection of *Fusarium solani* using a specific restriction site in the *EF-1a* gene [176]. *Fusarium solani* has been shown to have a worse prognosis, including higher voriconazole resistance, compared to other *Fusarium* species [177]. If rapid species identification using molecular methods were readily available, tailored treatment could be started earlier, thereby potentially improving the overall prognosis. However, the expense of molecular diagnostic methods precludes their use in many settings where FK is prevalent and further highlights the need for low-cost, point of care diagnostic tests which could be made more widely available.

Matrix-assisted laser desorption/ionization time of flight-mass spectrometry (MALDI-ToF MS) is a relatively novel rapid and reliable, high-throughput tool for the identification of microorganisms, allowing the identification of fungal isolates within minutes [178]. It also benefits from a fast turnaround time and low cost for consumables, making it potentially relevant to tertiary referral centres in LMICs where the burden of fungal keratitis is greatest. However, there are no published studies comparing MALDI-ToF MS to conventional methods for diagnosing fungal keratitis. One study has compared MALDI-ToF MS to conventional morphology and PCR sequencing which included one sample of *Aspergillus* keratitis which showed a good level of agreement between the different modalities [179]. There are a number of case reports and case series that explain how it is a useful tool in rapidly diagnosing FK, particularly for rare or unusual organisms [180,181,182,183,184].

#### 4.1.3. In Vivo Confocal Microscopy

Fungal culture can have a relatively low yield-studies report a sensitivity of up to 50% [185,186]. Growth may be slow; several days, even weeks; and identification complicated due to poor sporulation in vitro. Microscopy is very helpful but can have its limitations, particularly given the infection is often deep within the stroma making yield from corneal scrapings poor [167]. Early treatment (and therefore diagnosis) is crucial in treating FK appropriately and preventing the blinding complications associated with it [1]. A potential answer to these challenges comes in the form of in vivo confocal microscopy (IVCM), which allows for real-time imaging of the cornea down to the cellular and micro-structural level. It is able to detect the presence of fungal hyphae, Figure 8 [185,186].

IVCM can be used in the diagnosis of FK as well as in monitoring the response to treatment [185,186,187,188,189]. Chidambaram et al. reported a sensitivity of 79.1–86.8% and specificity of 73.7–85.9%, whilst Hau et al. correctly identified fungal infection 8.3–41.2% of the time [185,190]. However, it cannot reliably differentiate the organism causing the infection, meaning culture remains the gold standard for identification.

#### 4.1.4. Systematic Approach to Making a Diagnosis

With numerous tools available to aid in the diagnosis of fungal keratitis, it is useful to have a systemic approach. This will depend on what tools are available; as mentioned above, there are unfortunately many locations globally where access to these investigations are unavailable. In these locations, the algorithm in Figure 6 should be used. If all tests are available, we recommend following the algorithm given in Figure 9. A high index of suspicion is an important first step to diagnosing fungal infections: if a patient presents with a history of vegetative trauma, particularly if they are in a subtropical or tropical location, then fungal keratitis needs to be ruled out on the outset. As described above, if clinical signs including feathery or serrated margins, a raised profile or satellite lesions are present, then this should raise the probability of fungal keratitis. At this point a baseline corneal photograph is useful for future reference to guide future response, although staining with fluorescein should be delayed until after the PCR sample is taken to avoid theoretical interference with primers.

In these cases, the first investigation to be performed is IVCM. This should be performed before taking a corneal scrape, as taking a corneal scrape can reduce the image quality obtained by IVCM and therefore the sensitivity. Evidence of fungal hyphae are diagnostic. Ideally, the cornea should be anaesthetised with preservative free topical 0.5% proxymetacaine hydrochloride, as this is less likely to interfere with culture or PCR results. The subsequent step would be to take corneal scrapes for microscopy and culture, as described in detail in Section 4.1.1. It should be noted that a fresh sterile needle should be used for each slide or culture media being inoculated. Finally, a sample for PCR should be taken as a corneal swab. At this point, a second corneal photograph could be taken using a blue filter and topical fluorescein staining to demonstrate the size of the epithelial defect.

## 5. Management

Most cases of filamentous fungal keratitis are challenging to treat, requiring long-term therapy with topical, and occasionally systemic, antifungal agents. However, even when intensive appropriate topical therapy is initiated, infections frequently progress relentlessly to perforation and loss of the eye in ~25% of cases [1,22,24]. Surgery in the form of therapeutic penetrating keratoplasty (TPK) is often required. There are a limited number of antifungals available with action against fungal keratitis, of which there are four main groups: imidazoles, triazoles, polyenes and fluorinated pyrimidines. These may be available topically, orally or by intravenous injection. Subconjunctival injection or corneal stromal injection may also be given [27,28]. The current gold standard treatment for filamentous fungal keratitis is topical natamycin 5%.

There have been a number of clinical trials comparing various treatment options for fungal keratitis over the last few decades, which have been reviewed systematically [191,192]. Natamycin (NATA), which was approved in the 1960s by the FDA for FK, has been compared to a number of newer agents. In a randomised controlled superiority trial of 116 patients from India, there was no statistical difference found between econazole 2% or natamycin 5% [26]. Voriconazole, a newer generation triazole agent, was subsequently introduced to the market and an initial prospective RCT showed no significant difference between the groups in terms of primary outcome measure (time to healing of epithelial defect). The authors therefore concluded that voriconazole was “an effective and well-tolerated drug” and larger trials were warranted to demonstrate superiority [193]. Meanwhile, Prajna et al. also compared topical natamycin to voriconazole in a therapeutic exploratory randomised clinical trial; 120 patients were randomised to either natamycin or voriconazole and either had repeated corneal epithelial scraping or not. The study also concluded that there was no significant difference between groups for the primary outcome of visual acuity at three months, with a non-significant trend favouring voriconazole. Incidentally, repeated scraping was associated with a worse outcome, although again this was non-significant (*p* = 0.06) [194]. To investigate this, the Mycotic Ulcer Treatment Trials (MUTT) were developed [22,24]. In MUTT1, topical natamycin 5% was compared to topical voriconazole 1% in a trial that was due to recruit 368 patients but was terminated earlier on recommendation by the trial Data Safety and Monitoring Committee, as the number of perforations in the voriconazole group were significantly higher than in the natamycin group (34 vs. 18 perforations, *p* = 0.02; 323 recruited). Vision was −0.18 logMAR better at three months in the natamycin group compared to the voriconazole group (*p* = 0.006) [24]. Sharma et al. also found natamycin to be superior to voriconazole in a more recent randomised controlled trial [133].

MUTT 2 compared oral voriconazole with placebo with all patients receiving both natamycin and topical voriconazole. There was no difference in primary outcome (perforation rate or corneal graft) within three months between groups, with more side effects reported in the voriconazole group (*p* < 0.001). The study therefore concluded that there was no benefit in adding oral voriconazole in the treatment of severe filamentous fungal corneal infections [22]. As a result of these studies, topical natamycin 5% without oral voriconazole remains the recommended first-line agent for filamentous FK. MUTT also investigated the susceptibility of different fungal species to either medication, and found that *Aspergillus* spp. were least susceptible to natamycin, whilst *Fusarium* spp. were least susceptible to voriconazole. In the study population where MUTT was conducted, *Fusarium* spp. was the most commonly isolated organism. However, many patients continue to progress despite treatment with natamycin 5%, meaning that alternative treatment strategies are required. In addition, natamycin 5% is difficult to formulate, expensive and often unavailable in countries where it is required, despite being on the WHO Essential Medicines List. Chlorhexidine (CHX) is an antiseptic agent, with both antibacterial and antifungal properties. It is a widely used broad-spectrum biocide, killing microorganisms through cell membrane disruption. Pilot studies from the 1990s have suggested it as a potential alternative to natamycin 5% [23,25,191,195], and a randomised controlled trial comparing natamycin 5% to chlorhexidine 0.2% for fungal keratitis is currently underway [196].

Despite MUTT 2 showing no benefit for adjunctive oral voriconazole, some ophthalmologists recommend systemic, oral therapy in severe cases of fungal keratitis, particularly if the infiltrate is larger than 5 mm or deeper than 50% corneal thickness [197]. If oral voriconazole is not available, alternative options include ketoconazole or itraconazole. A randomised controlled trial comparing oral ketoconazole with oral voriconazole found similar healing times between groups, although patients treated with voriconazole achieved a significantly smaller scar size and better final vision [198]. It is, however, important to remember that these oral anti-fungal agents can have serious adverse effects, particularly in terms of hepatotoxicity; they should be used cautiously with correct dosing depending on the patient’s weight, together with liver function monitoring. Oral voriconazole has also been associated with treatment-related visual adverse events including blurred vision and colour vision changes [199], although these have been found to be non-progressive and reversible [199].

In addition to topical treatment, injections of antifungals into the corneal stroma have also been performed in severe disease [200,201]. This was investigated in a randomised controlled trial of 40 patients who were not responding to natamycin 5%, and compared topical voriconazole 1% to intrastromal injections of voriconazole 50 μg/0.1 mL. The authors found that patients receiving topical voriconazole had a mean BSCVA of −0.397 better than the intrastromal injection group (*p* = 0.008). Additionally, 19/20 patients receiving topical voriconazole healed with therapy. The authors concluded that topical, as opposed to intrastromal, voriconazole may be beneficial in addition to natamycin in recalcitrant disease not-responding to natamycin 5% monotherapy [202]. There is therefore no evidence indicating a benefit from intrastromal injections of voriconazole.

More recently, corneal collagen cross-linking (CXL) has been considered for the treatment of FK [203]. However, the evidence for this is limited with heterogenous protocols and conflicting results [204]. Indeed, three prospective randomised controlled trials have found no benefit of CXL over standard-of-care and, of concern, potentially worse outcomes in the CXL group [205,206].

The last intervention option for treating FK is surgical in the form of corneal transplantation or TPK. For large corneal perforations, TPK is the only option left to salvage the eye by restoring normal anatomy, with the added advantage of removing the site of the infection [207]. Unfortunately, however, recurrence of fungal infection in the graft often occurs, particularly in the presence of a hypopyon, corneal perforation, larger infiltrates and limbal involvement [208,209]. There is therefore a degree of debate around whether to perform TPK earlier in the course of the disease, rather than waiting for the eye to perforate, when future graft failure becomes more likely [207]. A recent retrospective study from India suggests that surgical intervention should be considered early in recalcitrant cases to improve the chances of graft survival [209]. However, TPK is a relatively technical procedure requiring an appropriately trained and experienced surgeon. Lack of donor graft availability is a significant challenge in large parts of the world where the need is greatest, in part due to legal and cultural barriers.

## 6. Ocular Mycology

### 6.1. Fusarium *spp.*

*Fusarium* keratitis is a sight-threatening condition that often affects otherwise healthy individuals during their most economically active years of life [1,210]. The infection is very challenging to treat due to resistance of *Fusarium* spp. to many antifungals. Without adequate treatment, infection progresses relentlessly to perforation [1,22,24], endophthalmitis [211], and ultimately loss of the eye in the form of enucleation [151,212].

#### Epidemiology

*Fusarium* keratitis is most common in tropical and sub-tropical locations [13]. The main risk factor for developing infection in this setting, in common with fungal keratitis with filamentous fungal aetiology, is trauma, typically with vegetative matter, resulting in a defect in the corneal epithelium [24,46,68,132,133,134]. This either directly inoculates the cornea with fungal conidia or allows subsequent fungal entry to the corneal stroma. There is a history of preceding trauma in 40–60% of cases [60,70]. Other risk factors include previous ocular surgery [151,152], ocular surface disease, previous use of corticosteroids [146], contact lens use [213], immunosuppression [146], or use of traditional eye medicines [142]. Fungal keratitis caused by *Fusarium* spp. accounts for between 42% and 52.5% of all cases of FK, depending on geographical location [14,28]. It typically occurs in young healthy males who are undertaking agricultural work [13].

However, *Fusarium* keratitis is not confined to the tropics. In tandem with the increased use of disposable planned-replacement contact lenses, the numbers of *Fusarium* keratitis reported in temperate countries with developed economies has also risen. As discussed above, between 2005 and 2006 there was an outbreak of contact lens-related *Fusarium* keratitis due to the contact lens cleaning solution “ReNu with MoisutureLoc” (Bausch & Lomb, Bridgewater, New Jersey, USA) [150]. The highest number of cases was seen in the Far East, with Hong Kong reporting 33 cases between January 2005 and May 2006 [214], and Singapore reporting *Fusarium* keratitis in 68 eyes of 66 patients between March 2005 and May 2006 [215]. Given the high prevalence of myopia in these industrialised locations and widespread, increasing use of soft contact lenses [216], it is not unsurprising that these countries saw the highest incidence during this outbreak. However, other countries including the USA (164 cases 2005–2006), [217] and European Nations reported a similar peak between 2005 and 2006 [27,218,219,220].

Irrespective of the ReNu outbreak, there appears to be an increasing incidence in *Fusarium* keratitis in temperate climates. In the UK, a London tertiary ophthalmic hospital reported an increase in the proportion of *Fusarium* spp. isolates of all fungal keratitis cases from 18% between 1994 and 2006 to 42% between 2007 and 2014 [28]. Contact-lens use was found to be a significant risk factor (OR 4.35, 95% CI 1.50–12.7). In Germany, the national reference laboratory have reported 15 cases of *Fusarium* keratitis over 2 years between January 2014 and December 2015 [10]. The majority of these were contact lens wearers (73.3%) with no cases reporting preceding trauma or immunosuppression. However, as the reference laboratory only commenced operations in 2014, comparisons to previous results was not possible. Similar reports of a rising incidence of *Fusarium* keratitis have been described from the Netherlands [11], which also finds contact lens use as a significant risk factor in this setting, as well as in Denmark where 9/10 cases were attributable to filamentous fungi between 2010–13, of which 6/9 were confirmed as *Fusarium* spp. [146]. Unlike *Fusarium* keratitis seen in tropical countries, in temperate climates it is more common in females, likely reflecting the demographics of contact lens use [10,11,28].

### 6.2. Aspergillus *sp.*

*Aspergillus* spp. are the second most frequently reported causative organisms of fungal keratitis globally. Several species have been associated with corneal infection, the commonest being *A. flavus*, *A. fumigatus*, *A. niger* and *A. terreus* [7,14,221]. Corneal trauma with vegetative or organic matter is the predominant risk factor reported [76]. The pattern of disease is similar to that seen with *Fusarium* keratitis, but in vitro susceptibility data for ocular isolates of *Aspergillus* spp. demonstrates lower MICs compared to antifungal susceptibility profiles for *Fusarium* spp. [221] although visual outcome is also determined by other factors such as the severity of the infection on presentation in clinic; deep lesions have a poorer prognosis [13].

#### Epidemiology

Mycotic keratitis due to *Aspergillus* spp. also predominates in tropical and sub-tropical latitudes [222]. However, within these regions and within countries there is climatic variation-wet, dry and semi-arid climes. *Aspergillus* corneal infections predominate in drier environments in sub-tropical latitudes, for example, in northern Ghana, where the environment is dry, with seasonal harmattan winds facilitating dispersal of airborne conidia; the more temperate areas of West Bengal and in northern India where the number of infections due to aspergilli eclipsed those caused by *Fusarium* spp., including in fungal keratitis in children [7,62,65,76,223].

### 6.3. Dematiaceous Fungi

The most commonly reported ocular pathogens after *Fusarium* spp. and *Aspergillus* spp. are representatives from the dematiaceous moulds, a diverse group of fungi characterised by their ability to produce melanin, which has long been regarded as a unique pathogenic advantage [224]. Although ubiquitous, this group of moulds are not common causes of disease in humans, but many species are plant pathogens of agricultural importance, colonising spoil and vegetation. The link with occupational risk factors and ocular trauma is as described for other types of mycotic keratitis.

Melanin pigmentation of hyphae and conidia within this heterogenous group may be useful in rapid diagnosis in this form of phaeohyphomycosis. Darkly pigmented infected corneal tissue may be obvious on direct observation of the eye, but this is not a common clinical presentation. There are few instances where morphological appearance of fungi in vivo are specific, however, direct microscopy of corneal tissue infected with some dematiaceous species may reveal pigmented fungal elements, including swollen, irregular hyphae and yeast-like structures, which are characteristic in appearance. Some species are weakly pigmented and may appear hyaline [225,226].

*Curvularia* spp. are the most commonly reported of the dematiaceous fungi globally. Many other genera have also been reported to cause keratitis including *Bipolaris* spp., *Exserohilum* spp., *Alternaria* spp., *Ulocladium* spp., *Lasidoplodia theobromae and Colletotrichum* spp. (Figure 10) [7,15,17,32,227].

#### Epidemiology

Ocular infections due to the dematiaceous fungi have been reported from every continent. Although more commonly reported from regions with warmer, humid seasonality members of this heterogenous group have also been reported from semi-arid regions [17,228,229,230,231,232]. In the terai of Nepal, the country with the highest documented incidence of fungal keratitis in the world, dematiaceous fungi such as *Curvularia* spp. are more frequently isolated than *Fusarium* spp. and *Aspergillus* spp., (personal experience & comms). *Curvularia* spp. were the most common filamentous fungi in a ten-year review of mycotic keratitis at a tertiary referral centre in North Carolina, South-eastern USA [122]. In order to understand regional patterns of causality it is important to reflect on the environmental reservoirs of many of these species, for example, *Curvularia* spp., which are pathogens of rice, maize, wheat, cassava, sorghum and grasses; common cash and subsistence crops in regions with a high incidence of fungal keratitis. To date there have been no phylogenetic studies comparing clinical (ocular) and environmental dematiaceous fungal isolates.

### 6.4. Other Filamentous Fungi

As previously mentioned there are more than 100 species of fungi reported as causing mycotic keratitis [13]. Other filamentous fungi less frequently reported include: *Sarocladium* spp., *Penicillium* spp., *Paecilomyces* spp., *Scedosporium* spp. and *Purepureocillum lilacinum*. Some of the least favourable therapeutic outcomes documented are mycotic keratitis cases due to *Scedosporium* spp., well characterised for their resistance to antifungal agents (Figure 11).

## 7. Unsolved Problems and Future Work

Fungal keratitis is a disease that disproportionately affects poor people living in some of the world’s poorest countries. There is evidence to suggest that the incidence of fungal keratitis is increasing globally. Unfortunately, for most people who have FK, access to appropriate diagnosis and treatment is very limited. To help address this apparent “neglect”, there has been a recent push for fungal keratitis, as part of microbial keratitis, to be included in the World Health Organization’s list of neglected tropical diseases (NTDs), which would help focus global attention and funding [233]. As it stands, there are a number of key areas where there are challenges and significant unmet needs, where addressing these may greatly reduce the morbidity associated with FK:Delay in presentation leading to poor outcomes [1,18]Use of traditional eye medicine and inappropriate use of conventional medicines [1,19,20]Limited relevant ophthalmic formal training of front-line health workers [1]Limited or no access to appropriate diagnostic investigationsTopical antifungals are frequently unavailable [1]FK is challenging to treat, and treatment failure is common [1,22,23,24,25,26]

Sight-loss from severe microbial keratitis (MK) in LMIC results from a combination of these factors. In response, current and future work is focused on addressing these areas. Research projects are underway to improve the understanding of patients’ health-seeking behaviour, such as that recently completed in Uganda [18,138,142]. Linked to this is implementation research into primary preventative measures, specifically how to prevent ocular injuries from occurring in the first place. Secondary preventative measures, for example antibiotic or antiseptic prophylaxis following corneal trauma, need to be enhanced. Several studies from South Asia found early antibiotic prophylaxis of uninfected corneal abrasions with chloramphenicol ointment reduced risk of MK developing [210,234,235,236]. However, these did not address early management of established MK presenting in the community, which still occurred in considerable numbers [210]. A suitable alternative to prevent fungal as opposed to bacterial keratitis also needs to be considered. Enhanced training of primary health workers, in addition with early referral, could potentially improve outcome.

To enhance the ability to accurately diagnose MK, microbiology laboratory capacity must be improved. This can be aided by the development of affordable point of care tests. As discussed above, the fungal species responsible (and therefore treatment susceptibility) varies with geographical location-and time-so it is essential for clinicians to be aware of the local aetiology to adjust treatment strategies. Continued microbiological surveillance is required to ensure that a change in aetiology is detected in good time.

Given that a large proportion of FK is attributed to trauma with vegetative material, and many fungal species causing FK are in fact plant pathogens, phylogenetic studies should be used to determine which plant pathogens are causing disease, specifically assessing their virulence and pathogenicity.

Regarding treatment, despite natamycin being added to the WHO Essential Medicines List in 2017 which was a huge step forward, there are still frequent shortages and in many countries is still not licensed or available [1]. When it is available, it is often too expensive for most people. Accessibility to the current gold standard treatment needs to be improved and the evidence-base into alternative treatments, such as chlorhexidine 0.2%, needs to be widened. Randomised controlled trials are currently underway to assess its efficacy [196].

## 8. Conclusions

Mycotic keratitis, particularly when caused by filamentous fungi, is a global problem. The incidence and main risk factors vary with geographical location and level of economic development; in tropical LMICs, trauma with organic material is the main risk factor whilst in wealthier, temperate countries contact lens use or ocular surface disease are the predominant associations. There is emerging evidence that the incidence is increasing worldwide, possibly linked in part to climate change, with other factors at play; further research is required to explore this in detail. Unfortunately, mycotic keratitis remains a severe, sight-threatening condition for millions.

## Figures and Tables

**Figure 1 jof-07-00273-f001:**
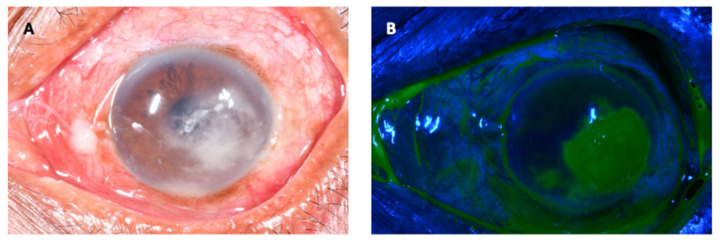
Fungal keratitis in a patient presenting to an ophthalmic hospital in Nepal. The causative organism was confirmed to be *Fusarium* sp. on culture. (**A**): The conjunctiva is hyperaemic, causing the eye to be red. There is a white corneal infiltrate with feathery serrated margins and satellite lesions present. There is also a small hypopyon. (**B**): The same eye as viewed with a cobalt blue filter after instillation of topical fluorescein. The area staining in green represents a defect in the corneal epithelium.

**Figure 2 jof-07-00273-f002:**
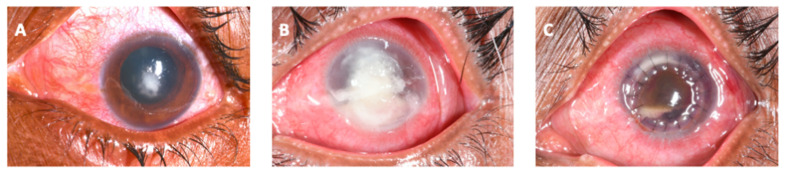
The progression of a patient with fungal keratitis caused by *Aspergillus* sp. This patient presented early in the course of the disease with a relatively small corneal ulcer, with serrated feathery margins to the corneal infiltrate (**A**). Despite intense, appropriate, prompt treatment with topical natamycin 5%, the corneal infiltrate increased in size, ultimately perforating, and was temporarily treated with corneal gluing and bandage contact lens insertion (**B**). The patient ultimately underwent a therapeutic penetrating keratoplasty (**C**).

**Figure 3 jof-07-00273-f003:**
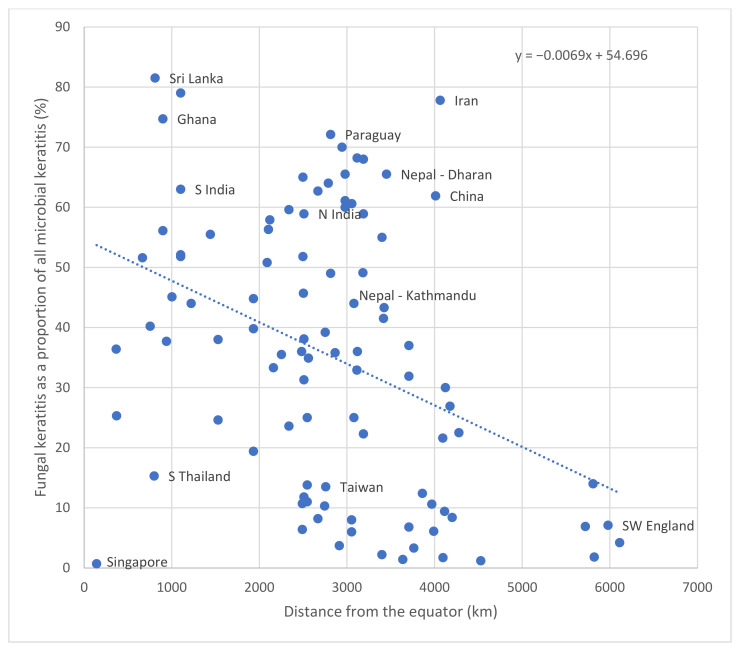
Fungal keratitis as a proportion of all culture positive microbial keratitis cases, by distance from the equator, with select locations shown, with calculated line of best fit given (dotted line, y = − 0.0069x + 54.696).

**Figure 4 jof-07-00273-f004:**
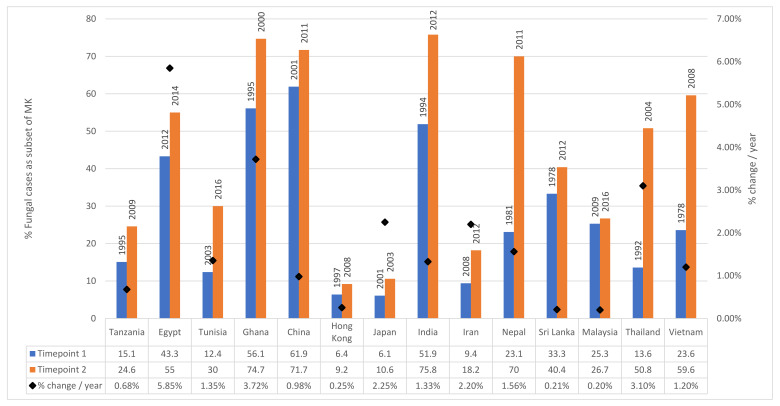
Percentage of fungal cases as a subset of MK plotted by country at two timepoints. Timepoint 1 represents the earliest year for which values were available, Timepoint 2 represents the latest year for which values are available. The years for the two studies are given as labels. The percentage change per year (calculated from the difference between the two timepoints) is plotted against the secondary y-axis.

**Figure 5 jof-07-00273-f005:**
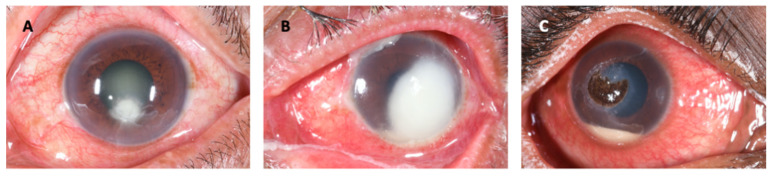
Differing clinical phenotypes of filamentous fungal keratitis depending on the fungal organism. (**A**): *Fusarium* sp. Note the serrated or feathery margins, satellite lesions, non-yellow infiltrate and lack of hypopyon. (**B**): *Aspergillus* sp. Note less obviously serrated margins compared to (**A**), raised profile, hypopyon. (**C**): *Curvularia* sp. Note the raised, pigmented infiltrate, in addition to the hypopyon.

**Figure 6 jof-07-00273-f006:**
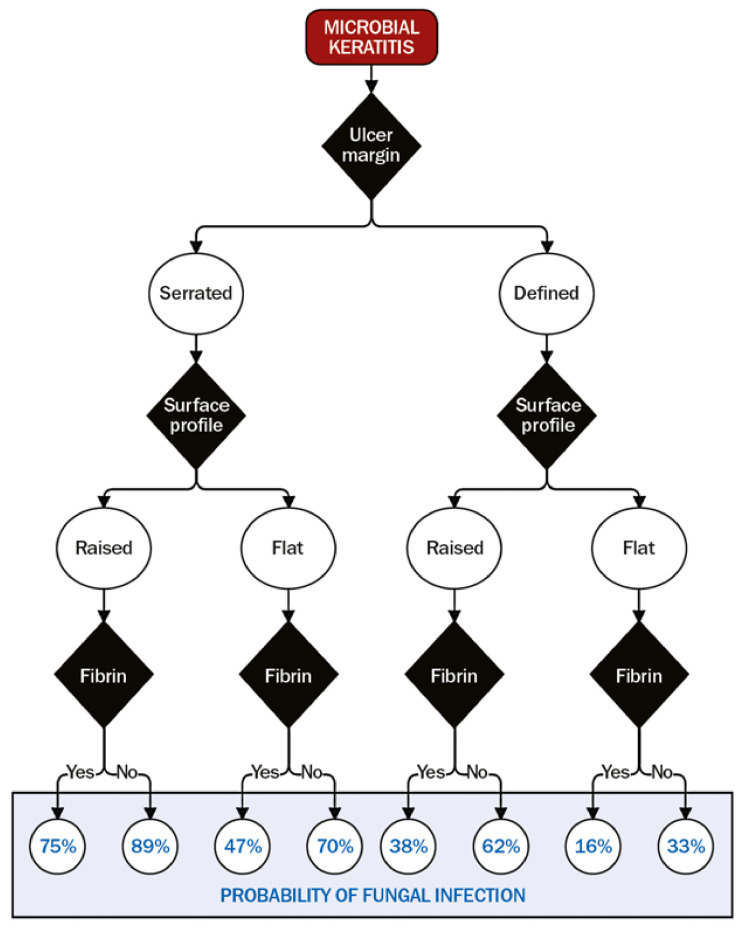
Algorithm for determining the probability of fungal keratitis [163]. The black diamonds are decision points about three clinical features: ulcer/infiltrate margin, surface profile, and anterior chamber fibrin. These probabilities are based on data presented in Thomas et al. [2]. This is reproduced here from [163] with permission under a CC BY-NC 4.0 license (https://creativecommons.org/licenses/by-nc/4.0/, accessed on 16 March 2021).

**Figure 7 jof-07-00273-f007:**
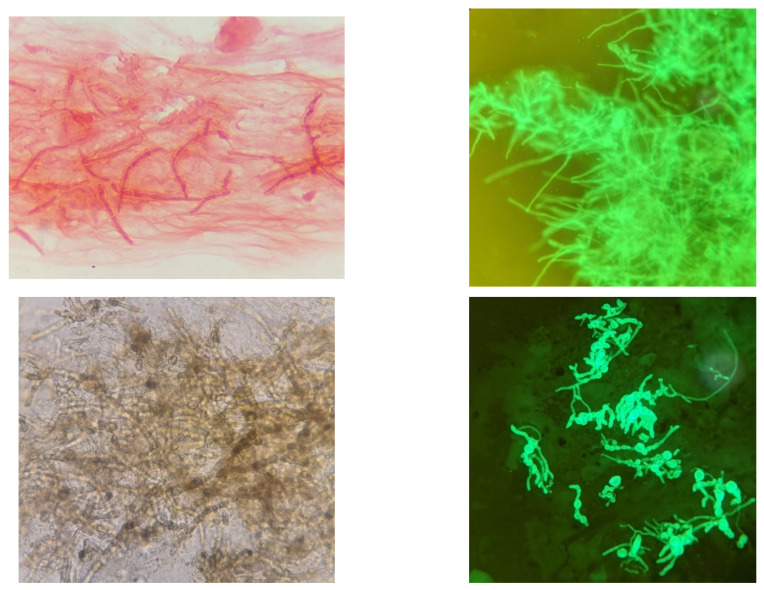
Microscopic appearance of filamentous fungal hyphae in corneal tissue (corneal scrape specimens) using different staining techniques. Clockwise from **top-left**: Fungal hyphae in Gram-stained corneal smear (magnification 1000x, oil immersion); fungal hyphae visible with CFW, *Curvularia* sp. stained with CFW, pigmented hyphae (*Curvularia* sp.) in a KOH preparation (magnification 400x). These images were taken using an afocal photography technique; the camera zoom was used for additional magnification.

**Figure 8 jof-07-00273-f008:**
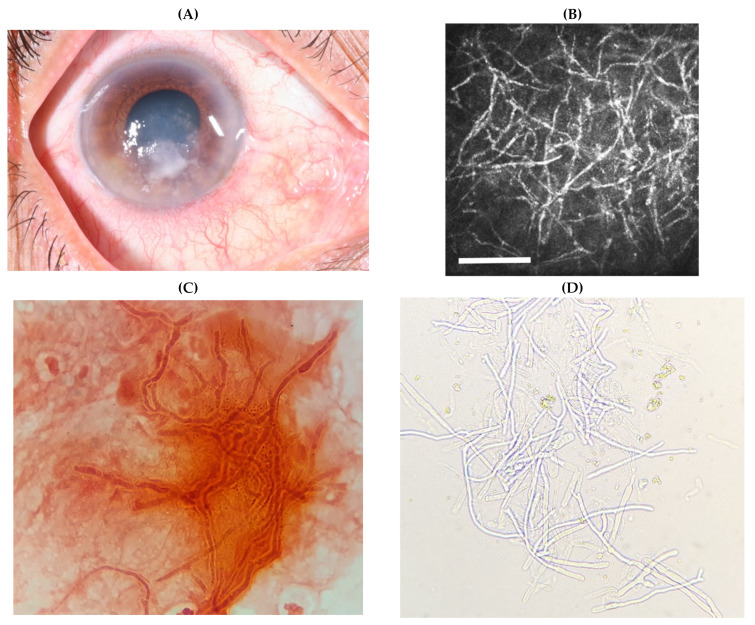
In vivo confocal microscopy of fungal keratitis (**A**) clinical image; (**B**) *In vivo* confocal microscopy scan of the same cornea showing extensive, branching fungal hyphae. Scale bar 100 μm. (**C**) Light microscopy demonstrated septate fungal hyphae, visible on Gram staining (magnification 1000x, oil immersion); (**D**) and KOH preparation (magnification 400x). Images (**C**,**D**) were taken using an afocal photography technique; the camera zoom was used for additional magnification.

**Figure 9 jof-07-00273-f009:**
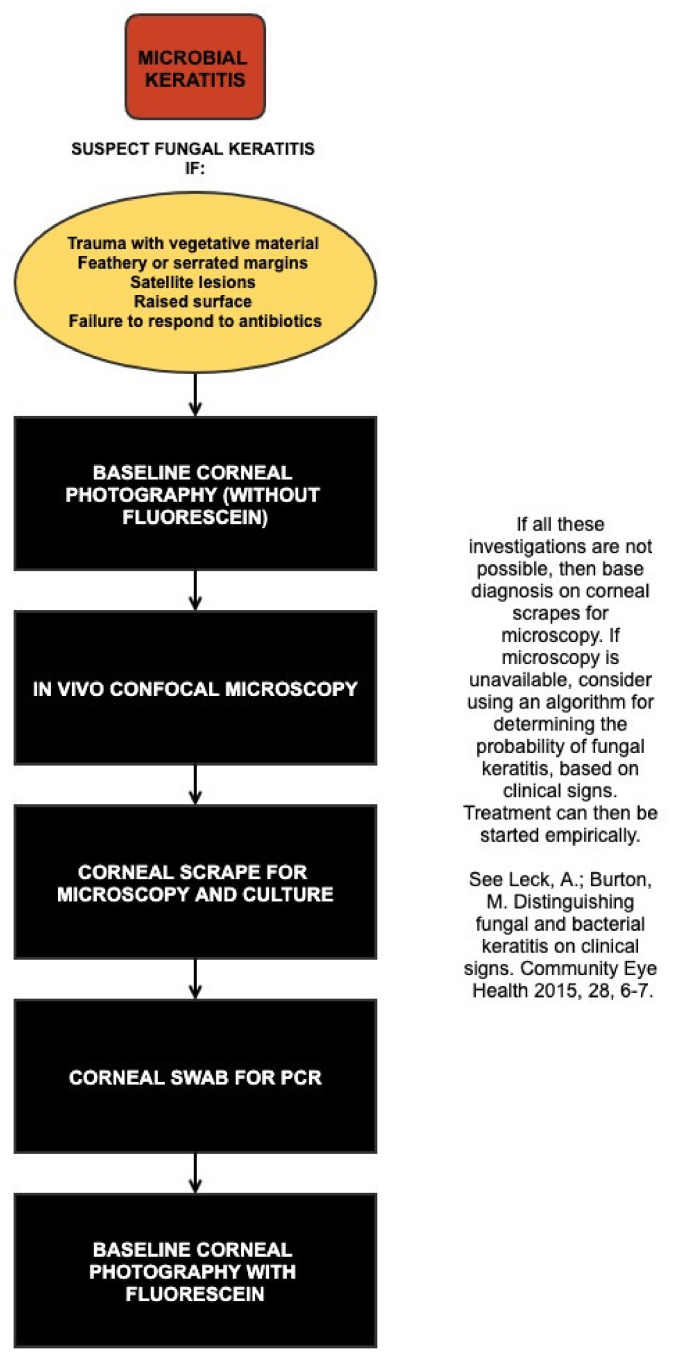
Algorithm for diagnosing fungal keratitis.

**Figure 10 jof-07-00273-f010:**
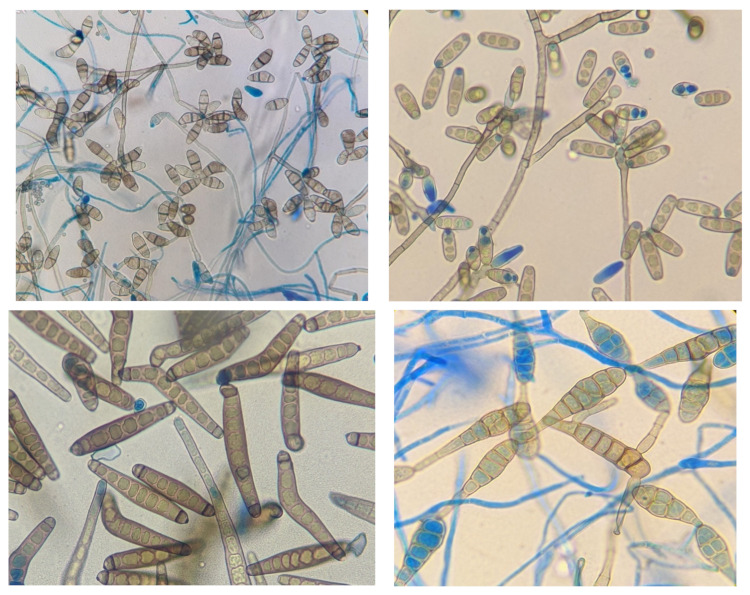
Examples of dematiaceous fungal genera isolated from cases of fungal keratitis stained with LPCB. Clockwise from **top-left**: *Curvularia* sp., *Bipolaris* sp. (magnification 400x); *Alternaria* sp., *Exserohilum* sp. (magnification x1000, oil immersion). These images were taken using an afocal photography technique; the camera zoom was used for additional magnification.

**Figure 11 jof-07-00273-f011:**
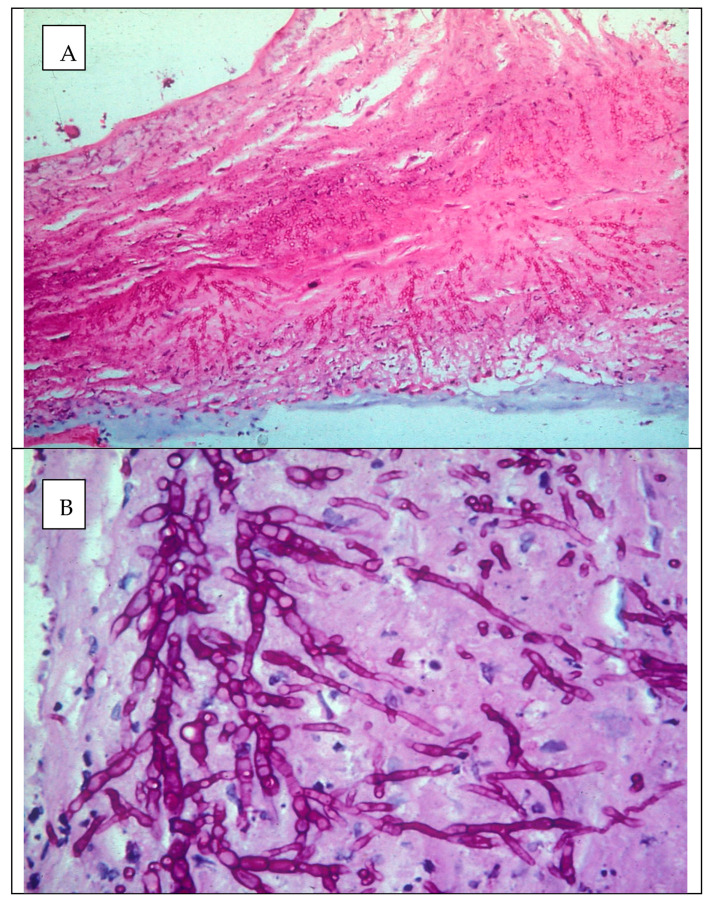
Histology section of a corneal button infected with *Scedosporium apiospermum* stained with H&E/PAS (A—magnification ×100, B—magnification ×1000, oil immersion).

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
