# Peer review of "Mycotic Keratitis—A Global Threat from the Filamentous Fungi"

_jof, 2021, doi:10.3390/jof7040273_

Round 1
Reviewer 1 Report
JOF keratitis.
The author of the review entitled “Mycotic keratitis – a global threat from the filamentous fungi” provide an extensive lit review of the different aspects of these infections. It’s a bit too long but mostly well written. The only item that was not cover is the incidence of these infections among the numerous patients receiving monthly injections for the treatment of retina problems. Nothing in le literature about that?
A few specific comments below:
Line 54, typo to correct.
Lines 56-60. Repetitive statement.
Lines 142-46. Perhaps, this incidence and the ones cited a couple of lines above could be summarized together? The proximity to the equator is becoming repetitive.
Confusing, what is “Africa” in the upper middle of the table?
The spp. next to the genus does not need to be in Italics, unless the journal demands it, please check the whole MS and correct accordingly.
Lines 212 to 235. Nothing about being caused by repetitive retinal treatments.
Lines 325 and on. What about the more frequent/invasive use of retina injections?
P359, are “serrated margin, raised slough” commonly known terms? Perhaps arrows could be helpful?
Photos and lines below or above the photos: Is there a need to make so much emphasis regarding the different stains when a simple KOH prep shows the hyphal elements clearly. Are all the photos needed?
Paragraph 503. One of the side effects of voriconazole could affect the eye,
Paragraph 6.1.1. Could this section be integrated in the prior sections where the therapy with antifungal agents was described?
Something off with ref.37
Author Response
The author of the review entitled “Mycotic keratitis – a global threat from the filamentous fungi” provide an extensive lit review of the different aspects of these infections. It’s a bit too long but mostly well written. The only item that was not cover is the incidence of these infections among the numerous patients receiving monthly injections for the treatment of retina problems. Nothing in le literature about that?
We would like to thank the reviewer for taking the time to review our manuscript. Thank you for highlighting the fact that we have not covered fungal keratitis for patients who are receiving regular intravitreal or subconjunctival injections for patients with retinal pathology. To the best of our knowledge at the time of writing this article, we did not consider intravitreal or peri-ocular injections to treat retinal disease as established risk factors for fungal corneal infections. In response to this comment, we have performed a specific literature search to ensure that this was not an area that we were missing. This search confirmed that there have not been any reported fungal corneal infections related to intravitreal or peri-ocular injections for patients with underlying retinal disease. However, there have been a number of reported cases of fungal endophthalmitis, fungal orbital abscesses or conjunctival mycetoma following intra-ocular or peri-ocular injections. These fungal infections, although important, do not fall within the scope of our article that specifically is reviewing fungal corneal infections.
However, we realised that we did not include previous ocular surgery as a potential risk factor for fungal keratitis. This is an established risk factor and we have therefore added this to Section 2.4 Risk Factors, including some select references. We have also added an explanation. This now reads:
2.4.10. Previous ocular surgery
A prior history of ocular surgery, including cataract, laser-refractive or corneal transplantation surgery, has been associated with the development of fungal keratitis in both developed and lower-middle income countries [152,153]. Yeasts are often the most commonly implicated pathogen following surgery [8]; for example, in a study from Boston, USA, yeasts accounted for 67% of post-surgical fungal infections. Of note, this group of patients had the worst outcome in terms of final visual acuity. In this study, all surgeries were a form of corneal transplantation [154]. However, it should be noted that prior ocular surgery is more likely to be a stronger risk factor for bacterial, rather than fungal, keratitis [61]; a study from Brazil found 32% of bacterial keratitis cases were associated with previous ocular surgery, compared to just 8% of fungal keratitis cases [155].
Despite intravitreal injections for retinal disease becoming the most commonly performed intraocular procedure globally [156], and corticosteroid periocular injections being used routinely for the treatment of diabetic macular oedema [157,158], there have been no cases of fungal keratitis associated with this treatment reported in the scientific literature to date. However, other complicating local fungal infections have been reported, including fungal endophthalmitis, fungal orbital abscesses and conjunctival mycetoma [159-161].
A few specific comments below:
Line 54, typo to correct.
Thank you, we have reviewed this line but were unable to see a typo. Please can you clarify what the misspelled word is? For reference, we have chosen to use the word “clime” as opposed to “climatic region” or similar for its brevity and hope this is acceptable.
Lines 56-60. Repetitive statement.
Thank you for this comment. We have clarified this statement as we feel that it is not repetitive, as it describes how poor people in rich countries seem to adversely suffer from corneal infections. It was, however, a little ambiguous and so we have rephrased this section so that it now reads:
There have also been reports of an increase in Fusarium-related keratitis in contact lens wearers in temperate, industrialised regions [9-11]. Interestingly, even within developed countries fungal keratitis is a disease of poverty: infections are associated with contact lens wearers from deprived or low socioeconomic backgrounds [3,12].
Lines 142-46. Perhaps, this incidence and the ones cited a couple of lines above could be summarized together? The proximity to the equator is becoming repetitive.
Thank you for this comment. We have made a few minor changes to this paragraph and removed some text to try to rephrase and reduce the repetitiveness. This section now reads:
The first study from 2002 plotted the proportion of FK as a subset of MK against latitude, and found the proportion of FK cases increases with decreasing latitude, i.e. increasing the closer one is to the equator [7]. The second review, from 2011, correlated the proportion of fungal cases of MK in a country with the country’s gross domestic product (GDP) [31]. This found the highest proportion of fungal infections within Asia, specifically in India and Nepal. The study found the lower the GDP per capita of a country, the higher the proportion of fungal MK. The most recent study looked at both GDP per capita and latitude as potential determinants of the proportion of fungal cases of all those with MK [3]. The findings here correlated to the previous two reviews, suggesting that both proximity to the equator and low GDP per capita are associated with a higher proportion of fungal MK cases [3,7,31]. However, it is important to note there was some considerable unexplained variability [3].
Confusing, what is “Africa” in the upper middle of the table?
Thank you for this comment. I have updated the table legend to explain this. This now reads as follows:
Table 1: Global epidemiology of fungal keratitis (FK), most frequently isolated fungal organisms and summary results of select papers on risk factors for developing FK, grouped as per their geographical region (as defined by the UN).
The spp. next to the genus does not need to be in Italics, unless the journal demands it, please check the whole MS and correct accordingly.
Noted. This has been corrected, thank you.
Lines 212 to 235. Nothing about being caused by repetitive retinal treatments.
Thank you. Please see my response to your general comment above. I have added the following to the text here to address this query:
Despite intravitreal injections for retinal disease becoming the most commonly performed intraocular procedure globally [156], and corticosteroid periocular injections being used routinely for the treatment of diabetic macular oedema [157,158], there have been no cases of fungal keratitis associated with this treatment reported in the scientific literature to date. However, other complicating local fungal infections have been reported, including fungal endophthalmitis, fungal orbital abscesses and conjunctival mycetoma [159-161].
Lines 325 and on. What about the more frequent/invasive use of retina injections?
Please see the answer above.
P359, are “serrated margin, raised slough” commonly known terms? Perhaps arrows could be helpful?
Thank you for this comment. We have added an explanation within the text which should clarify this. The figures as they are are a little small to add an arrow clearly; however, if it is felt this would help further clarify this then an enlarged figure could be used with an arrow added. Please advise if you feel this would help strengthen and clarify the article.
Photos and lines below or above the photos: Is there a need to make so much emphasis regarding the different stains when a simple KOH prep shows the hyphal elements clearly. Are all the photos needed?
Thank you. We felt it is helpful for the reader to see how each preparation differs as different centres may have different preparations available, or be interested to try alternative options, and these images may help them in this regard.
Paragraph 503. One of the side effects of voriconazole could affect the eye,
Thank you, we have added the following that describes ocular adverse events with the use of oral voriconazole:
Oral voriconazole has also been associated with treatment-related visual adverse events including blurred vision and colour vision changes [200], although these have been found to be non-progressive and reversible [200].
Paragraph 6.1.1. Could this section be integrated in the prior sections where the therapy with antifungal agents was described?
Thank you for this comment. 6.1.1 relates to the epidemiology of Fusarium keratitis. We have not found a way to make this fit within the previous sections. Perhaps the reviewer may have been referring to a different paragraph or section? If so, please advise and we will try and amend accordingly.
Something off with ref.37
Thank you, the formatting for this has been corrected.
Reviewer 2 Report
In the current review “Mycotic keratitis – a global threat from the filamentous fungi” the authors provided an in-depth review of microbial keratitis. Authors elaborated the on geographical location and the various microbial pathogen causing the corneal infection. Overall, the article is well written and provides new information as well as challenges in the diagnosis of microbial keratitis.
Here we are giving comments/suggestions to improve the current review.
1) Authors should provide the tabulated or diagrammatic representation of the diagnostic test available. It will help the readers to see and understand the summary of various basic, molecular biology tests.
2) The authors mentioned that corticosteroids are one of the risk factors, so what anti-inflammatory therapies are recommended?
3) Authors mentioned that there is an increased incidence of fungal keratitis, and provided various justifications like an increase in temperature, contact lens usage, etc. Is it also possible, that previously these cases were underreported? Also, is there any chance of fungal keratitis with an increased frequency of intravitreal injections? Is there any such study or correlation data available? If appropriate authors can include these points in the review.
4) What is the impact of fungal keratitis on visual functions? If visual functions declines are these reversible with the treatment?
Author Response
In the current review “Mycotic keratitis – a global threat from the filamentous fungi” the authors provided an in-depth review of microbial keratitis. Authors elaborated the on geographical location and the various microbial pathogen causing the corneal infection. Overall, the article is well written and provides new information as well as challenges in the diagnosis of microbial keratitis.
Thank you for taking the time to review our manuscript and for your constructive comments.
Here we are giving comments/suggestions to improve the current review.
1) Authors should provide the tabulated or diagrammatic representation of the diagnostic test available. It will help the readers to see and understand the summary of various basic, molecular biology tests.
Thank you for this comment, we have added the following section and diagram to help clarify the steps to making a diagnosis:
4.1.4 Systematic approach to making a diagnosis
With numerous tools available to aid in the diagnosis of fungal keratitis, it is useful to have a systemic approach. This will depend on what tools are available; as mentioned above, there are unfortunately many locations globally where access to these investigations are unavailable. In these locations, the algorithm in Figure 6 should be used. If all tests are available, we recommend following the algorithm given in Figure 9. A high index of suspicion is an important first step to diagnosing fungal infections: if a patient presents with a history of vegetative trauma, particularly if they are in a subtropical or tropical location, then fungal keratitis needs to be ruled out on the outset. As described above, if clinical signs including feathery or serrated margins, a raised profile or satellite lesions are present, then this should raise the probability of fungal keratitis. At this point a baseline corneal photograph is useful for future reference to guide future response, although staining with fluorescein should be delayed until after the PCR sample is taken to avoid theoretical interference with primers.
In these cases, the first investigation to be performed is IVCM. This should be performed before taking a corneal scrape, as taking a corneal scrape can reduce the image quality obtained by IVCM and therefore the sensitivity. Evidence of fungal hyphae are diagnostic. Ideally, the cornea should be anaesthetised with preservative free topical 0.5% proxymetacaine hydrochloride, as this is less likely to interfere with culture or PCR results. The subsequent step would be to take corneal scrapes for microscopy and culture, as described in detail in Section 4.1.1. It should be noted that a fresh sterile needle should be used for each slide or culture media being inoculated. Finally, a sample for PCR should be taken as a corneal swab. At this point, a second corneal photograph could be taken using a blue filter and topical fluorescein staining to demonstrate the size of the epithelial defect.
Figure 9: Algorithm for diagnosing fungal keratitis.
2) The authors mentioned that corticosteroids are one of the risk factors, so what anti-inflammatory therapies are recommended?
Thank you for this comment. Unfortunately, there are no safe or effective topical anti-inflammatory agents currently used or recommended for the treatment of fungal keratitis. We have therefore not mentioned this within the article. If you feel the article would benefit from this being mentioned, then please let us know and we can add to this section or elsewhere.
3) Authors mentioned that there is an increased incidence of fungal keratitis, and provided various justifications like an increase in temperature, contact lens usage, etc. Is it also possible, that previously these cases were underreported? Also, is there any chance of fungal keratitis with an increased frequency of intravitreal injections? Is there any such study or correlation data available? If appropriate authors can include these points in the review.
Thank you for this comment. Regarding a possible link with intravitreal injections, we have performed a further search of the literature and found no such link. Interestingly, Reviewer 1 also raised this question. We have therefore explained this now within the text within a section that describes ocular surgery as a risk factor. This section reads as follows:
2.4.10. Previous ocular surgery
A prior history of ocular surgery, including cataract, laser-refractive or corneal transplantation surgery, has been associated with the development of fungal keratitis in both developed and lower-middle income countries [152,153]. Yeasts are often the most commonly implicated pathogen following surgery [8]; for example, in a study from Boston, USA, yeasts accounted for 67% of post-surgical fungal infections. Of note, this group of patients had the worst outcome in terms of final visual acuity. In this study, all surgeries were a form of corneal transplantation [154]. However, it should be noted that prior ocular surgery is more likely to be a stronger risk factor for bacterial, rather than fungal, keratitis [61]; a study from Brazil found 32% of bacterial keratitis cases were associated with previous ocular surgery, compared to just 8% of fungal keratitis cases [155].
Despite intravitreal injections for retinal disease becoming the most commonly performed intraocular procedure globally [156], and corticosteroid periocular injections being used routinely for the treatment of diabetic macular oedema [157,158], there have been no cases of fungal keratitis associated with this treatment reported in the scientific literature to date. However, other complicating local fungal infections have been reported, including fungal endophthalmitis, fungal orbital abscesses and conjunctival mycetoma [159-161].
Regarding the under-reporting of fungal keratitis previously, yes, we agree that indeed there is a potential argument that fungal keratitis was previously under-reported by inadequate culture and microscopy. This was mentioned in the text here, although this may not have been very clear. We have therefore amended this section to read as follows and hope that this clarifies this section:
Other potential reasons include increased availability and use of topical steroids, increased prevalence of diabetes mellitus across the regions or simply due to improved culture and microbiology services in these countries, meaning that under-reported previous incidence is now being reported more accurately. Increased contact lens wear may also be a contributing factor, although on the whole contact-lens use remains infrequent in poorer countries across Asia and Africa.
4) What is the impact of fungal keratitis on visual functions? If visual functions declines are these reversible with the treatment?
Thank you for this comment. In answer to this question, we have added the following paragraph to Section 3 – Clinical Features (it could also feature in the introduction, please advise if you think it would be better suited there):
Acutely, fungal keratitis typically leads to reduced vision due to the presence of the infection and inflammation in the cornea, blurring the vision. With treatment, the vision can improve, although often the patient is left with worse vision than they had previously due to the development of corneal scarring. At present there is no medical treatment to reverse this scarring process. Rigid contact lenses can help to a certain amount by improving the vision if there is scarring. Alternative options for severe scarring include corneal graft surgery, but this can be a technically challenging procedure and is often not available in places most in need. Fungal keratitis should therefore be considered a potentially blinding condition.